# GABA and Fermented *Curcuma longa* L. Extract Enriched with GABA Ameliorate Obesity through Nox4-IRE1α Sulfonation-RIDD-SIRT1 Decay Axis in High-Fat Diet-Induced Obese Mice

**DOI:** 10.3390/nu14081680

**Published:** 2022-04-18

**Authors:** Hwa-Young Lee, Geum-Hwa Lee, The-Hiep Hoang, Yu-Mi Kim, Gi-Hyun Jang, Chang-Hwan Seok, Yun-Geum-Sang Gwak, Junghyun Lim, Junghyun Kim, Han-Jung Chae

**Affiliations:** 1Non-Clinical Evaluation Center Biomedical Research Institute, Jeonbuk National University Hospital, Jeonju 54907, Jeollabuk-do, Korea; youngat84@gmail.com (H.-Y.L.); heloin@jbnu.ac.kr (G.-H.L.); drhiep.ydhue@gmail.com (T.-H.H.); 2Research Institute of Clinical Medicine of Jeonbuk National University, Biomedical Research Institute of Jeonbuk National University Hospital, Jeonju 54907, Jeollabuk-do, Korea; 3Binotec Co., Ltd., 155 Deulan-ro, Suseong-gu, Daegu 41566, Gyeongsangbuk-do, Korea; yandme77@binotec.co.kr (Y.-M.K.); ghjang@binotec.co.kr (G.-H.J.); seokch1114@binotec.co.kr (C.-H.S.); goldprize95@binotec.co.kr (Y.-G.-S.G.); 4School of Pharmacy, Jeonbuk National University, Jeonju 54896, Jeollabuk-do, Korea; jl1206@jbnu.ac.kr; 5Department of Oral Pathology, School of Dentistry, Jeonbuk National University, Jeonju 54896, Jeollabuk-do, Korea; dvmhyun@jbnu.ac.kr

**Keywords:** NADPH oxidase, IRE1α, SIRT1, decay, obesity

## Abstract

*Gamma*-*aminobutyric acid* (GABA) is a natural amino acid with antioxidant activity and is often considered to have therapeutic potential against obesity. Obesity has long been linked to ROS and ER stress, but the effect of GABA on the ROS-associated ER stress axis has not been thoroughly explored. Thus, in this study, the effect of GABA and fermented *Curcuma longa* L. extract enriched with GABA (FCLL-GABA) on the ROS-related ER stress axis and inositol-requiring transmembrane kinase/endoribonuclease 1α (IRE1α) sulfonation were examined with the HFD model to determine the underlying anti-obesity mechanism. Here, GABA and FCLL-GABA supplementations significantly inhibited the weight gain in HFD fed mice. The GABA and FCLL-GABA supplementation lowered the expressions of adipogenic transcription factors such as PPAR-γ, C/EBPα, FAS, and SREBP-1c in white adipose tissue (WAT) and liver from HFD-fed mice. The enhanced hyper-nutrient dysmetabolism-based NADPH oxidase (Nox) 4 and the resultant IRE1α sulfonation-RIDD-SIRT1 decay under HFD conditions were controlled with GABA and FCLL-GABA. Notably, GABA and FCLL-GABA administration significantly increased AMPK and sirtuin 1 (SIRT1) levels in WAT of HFD-fed mice. These significant observations indicate that ER-localized Nox4-induced IRE1α sulfonation results in the decay of SIRT1 as a novel mechanism behind the positive implications of GABA on obesity. Moreover, the investigation lays a firm foundation for the development of FCLL-GABA as a functional ingredient.

## 1. Introduction

Obesity is a growing global health concern associated with multiple chronic diseases, such as type 2 diabetes, multiple cancers, cardiovascular diseases, and hypertension [1]. White adipose tissue (WAT) plays a crucial role in regulating systemic energy and is essential for the proper functioning of several metabolic tasks to manage obesity-associated chronic diseases. However, it loses its functional capabilities with developing obesity as it impairs WAT’s abilities to store surplus energy. This leads to ectopic fat accumulation in various tissues that influence metabolic homeostasis [2]. Hence, therapeutic initiatives targeting lipogenesis and lipid dysmetabolism in obesity and linked clinical conditions have potential therapeutic applications [3].

During WAT-associated metabolic processes, the NADPH oxidase and its linked superoxide need to be tightly controlled for metabolic homeostasis. An increase in the activity of NADPH oxidase enhances the production of superoxides leading to metabolic disorders due to increased oxidative stress. NADPH oxidase-associated reactive oxygen species (ROS) are assumed to be the cause of the reduced insulin responsiveness [4,5], endoplasmic reticulum (ER) dysfunction, and ROS production in the ER [6,7]. During stress, protein dysfunction is influenced by sulfonation, which involves the oxidation of specific cysteine residues mediated by endogenous or exogenous ROS [8]. Moreover, ER-derived H_2_O_2_ is produced from Nox4 and leads to ER stress [9]. Inositol-requiring enzyme 1α (IRE1α) sulfonation is activated via the NADPH oxidase 4 (Nox4)-ER-ROS axis and is linked to cellular dysfunction [10]. The IRE1α post-translational modification (PTM)-linked signaling axis, known as regulated IRE1-dependent decay (RIDD), has diverse substrates, including proteins associated with glucose metabolism. Moreover, sirtuin (SIRT) family genes played critical roles in metabolic disorders. Specifically, ER stress axis-linked SIRT is a frequently investigated pathway, but the application of the IRE-1-RIDD axis to SIRT’s fate is yet to be defined.

*Gamma*-*aminobutyric acid* (GABA) is an amino acid that serves as an inhibitory neurotransmitter for the central nervous system (CNS). Several reports have identified GABA as an anti-obesity compound that regulates obesity through anti-inflammatory, antioxidant, and other properties that improve glucose metabolism [11,12,13,14,15]. In a high-fat diet (HFD) model, supplementation of GABA suppressed weight gain and reduced fat mass [12,14,15]. Similarly, administration of GABA in lean and diabetic mice revealed positive implications on glucose metabolism, with an increase in β-cell mass and function, without affecting the bodyweight [13,16,17]. Moreover, clinical trials suggest that GABA potentially enhances circulating insulin and glucagon [18,19]. However, GABA therapeutic applications were restricted by potential side effects, including upset stomach, headache, sleepiness, and muscle weakness. Hence, GABA-enriched natural products such as fermented food or herbal medicine have been proposed as a promising alternative strategy to treat metabolic disorders. For ages, fermentation has been more popular and relatively simple than chemical/biotechnological processes. Thus, GABA-enriched fermented foods might be the best choice to replace GABA. Further, accumulated investigations with *Curcuma longa* L. on obesity prompted us to evaluate the effects of fermented *Curcuma longa* L. extract enriched with GABA (FCLL-GABA). 

In this study, the effects of GABA and fermented *Curcuma longa* L. extract enriched with GABA on obesity were evaluated using the HFD model and an effort was made to extrapolate the mechanism responsible for its anti-obesity effect. 

## 2. Materials and Methods

### 2.1. Preparations of GABA and Fermented Curcuma longa L. Extract Enriched with GABA

Fermented *Curcuma longa* L. extract enriched with GABA was produced as suggested previously with minor modifications [20]. The fermented *Curcuma longa* L. extract enriched with GABA (FCCL), *Hydroiet™,* was provided by Binotec Co., Ltd. (Daegu, Republic of Korea).

### 2.2. High Performance Liquid Chromatography (HPLC) Analysis

The Elite Lachrom HPLC-DAD system with UV (DAD) detector (Hitachi High-Technologies Co., Tokyo, Japan) and CAPCELL PAK C18 (UG120, 4.6 × 250 mm 5 μm) column was used to perform HPLC analysis. The column temperature was 40 °C, and the injection volume was 10 μL. The mobile phases were composed of the solvents acetonitrile (CAN), 10 mM phosphoric buffer (pH 2.4), and methanol. The run time was 25 min and the mobile phase process gradient flow was as follows: (A)/(B)/(C) = 1.5/97/1.5 (0 min) → (A)/(B)/(C) = 45//10/45 (0–25 min). The mobile phase flow rate was 1.0 mL/min, and the wavelength of the UV (DAD) detector was set at 280 nm.

### 2.3. Animal Studies

Male C57BL/6J mice aged 6 weeks were procured from Orient Science Co. (Seongnam, Korea). All the mice were maintained at 22 ± 2 °C with a standard L-D cycle. Prior to the experiment, all the animals were acclimatized for a week. Mice (7-wk old) were randomly selected and divided into six groups, each with eight mice. Group 1 (vehicle) normal chow diet (NCD) mice received water. Group 2 (vehicle) high-fat-diet (HFD) fed mice received water. Group 3 and 4 (FCLL-1 and FCLL-2) HFD fed mice received 1 and 2 g/kg fermented *Curcuma Longa* L. enriched with GABA (FCLL-GABA), respectively. Groups 5 and 6 (GABA-1 and GABA-2) HFD fed mice received 1 and 2 g/kg GABA, respectively. Reagents (FCLL and GABA in water) and vehicle (water) were administered daily via oral gavage. The experimental diet was followed up for 14 weeks, and at the end of the 14th week, all the mice were sacrificed to collect relevant samples. Tissue samples were stored below −80 °C, whereas whole blood samples were collected using EDTA tubes and stored at 2 °C for 30 min. Serum was obtained by centrifuging the whole blood at 3000× *g* for 10 min at 4 °C. Animal experiments were carried out in compliance with the guidelines set by the Jeonbuk National University hospital animal care and use committee (JBUH-IACUC-2021-4-1).

### 2.4. Biochemical Analysis

All the biochemical analyses were performed with commercial kits. Aspartate aminotransferase (AST, AM101-1), alanine aminotransferase (ALT, AM101-2), triglyceride (TG, AM1575K), and total-cholesterol (AM202K) levels in serum were analyzed using commercial kits from Asan Pharm, Co., Seoul, Korea by following guidelines. Adiponectin and leptin levels in plasma were evaluated using commercial sandwich ELISA kits (CSB-E07272m, CSB-E04650m, CUSABIO, USA). The absorbance was measured at 450 nm with a microplate reader (Multiskan SkyHigh; Thermo Fisher Scientific, Inc.). The calculations were conducted as per the manufacturer’s guidelines, and concentrations were reported in ng/mL.

### 2.5. Immunoblot Analysis

Immunoblotting was carried out as suggested previously [21]. All the immunoblots were probed with the appropriate antibodies. The following antibodies were used in this investigation: AMP-activated kinase (AMPK, #2532, cell signaling, MA, USA), phosphorylation of AMP-activated kinase (p-AMPK, P-2535, cell signaling, MA, USA), sterol regulatory element-binding protein (SREBP-1c, sc-36553, Santa Cruz, CA, USA), peroxisome proliferator-activated receptor γ (PPAR-γ, sc-7273, Santa Cruz, CA, USA), CCAAT/enhancer-binding protein α (C/EBP1α, sc-166258, Santa Cruz, CA, USA), fatty acid synthase (FAS, sc-74540, Santa Cruz, CA, USA), Nox4 (#PA5-72816, Thermo Fisher Scientific, Waltham, MA, USA), sirtuin 1 (SIRT1, sc-74465, Santa Cruz CA, USA), p-IRE1α (#ab124945, Abcam), GRP78 (#13539, Santa Cruz, CA, USA), CHOP (#2895, cell signaling, MA, USA), and β-actin (sc-47778, Santa Cruz, CA, USA). All the blots were probed with species-specific secondary antibodies, and signals were detected using enhanced chemiluminescent (ECL) reagent.

### 2.6. Histological Analysis and Immunohistochemistry (IHC)

Histological analysis and IHC were performed as described earlier [21]. Briefly, dissected WATs were fixed with 10% formalin, and paraffin-embedded samples were cut into 4 µm sections. Later, 4 µm sections were stained with H&E. Similarly, formalin-fixed and sectioned tissues were processed as reported previously [21].

### 2.7. Immunofluorescence Analysis

The eWAT (epididymal WAT) was fixed and washed with phosphate-buffered saline (PBS). The sections were exposed to a primary antibody against UCP-1 (sc-518024, Santa Cruz Biotechnologies Inc., CA, USA) for 2 h at room temperature. Following incubation with a primary antibody, the sections were washed thrice for 5 min in PBS. Later, the sections were labeled using anti-mouse IgG- FITC (Sigma, St Louis, MO, USA) for 1 h at room temperature. The sections were washed thrice for 5 min in PBS, and glass coverslips were mounted with 20 μL aqueous-mount solution (Scytek laboratories, Logan, UT, USA). Images were captured using an inverted confocal microscope (Leica DMIRE2; Leica Microsystems, Wetzlar, Germany) with a 63× oil immersion objective lens. All the images were captured with the same laser intensities.

### 2.8. DHE Staining Analysis

Dihydroethidium (DHE) was employed to analyze the intracellular ROS production in WAT. Fresh WATs were fixed in 4% paraformaldehyde on ice for 1 h. The fixed tissues were washed three times by PBS for 10 min on ice before dehydrating overnight in 30% sucrose at 4 °C. Then, the tissues were infiltrated with OCT (SAKURA, Torrance, CA, USA) for 2 h and stored at −80 °C. Sections (5 µm) were cut with a freezing microtome (CM3050S; Leica Microsystems, Wetzlar, Germany). The sections were dried at room temperature for 5 min and then washed three times by PBS for 5 min. Tissues were exposed to DHE (50 µM, diluted by PBS) at 37 °C for 30 min. All the images were acquired with EVOS M5000 Cell Imaging System (Thermo Fisher Scientific, Waltham, MA, USA), and relative fluorescence intensities were assessed with ImageJ (National Institutes of Health, Bethesda, MD, USA).

### 2.9. IRE1α Sulfonation Assay

Sulfonation was evaluated as described previously [22]. About ~500 μg of WAT were used to obtain WAT lysates. For detection of sulfonation of IRE1α, immunoprecipitation was performed with anti-cysteine-sulfonate (Abcam, ab176487) using lysates and incubated with anti-IRE1α antibody (3294, cell signaling) to detect IRE1α sulfonation. Later, protein A/G Sepharose beads were mixed and incubated for an hour at room temperature. Finally, immunoprecipitates were washed with phosphate-buffered saline (PBS), resolved using SDS-PAGE, and immunoblotted with appropriate antibodies.

### 2.10. NADPH-Dependent Oxidoreductase (Nox) Activity Assay

NADPH oxidase activity was assessed by analyzing superoxide generation through lucigenin-enhanced chemiluminescence (LECL) [23]. Washed WAT was homogenized with lysis buffer having 20 mM sodium phosphate buffer, 1 mM EDTA, 1 mM PMSF, 0.5 mM leupeptin, and 0.5 mM pepstatin. The reaction mixture was prepared using homogenates (5 µL) and 0.5 mL of assay buffer. Finally, 0.1 mM NADPH was added, and luminescence was measured using Skanlt RE 6.1 (Thermo Fisher Scientific, Waltham, MA, USA), a microplate reader. All the observations were expressed as RLU/mg of protein.

### 2.11. In Vitro IRE1α-Mediated SIRT1 Cleavage Assay

Cleavage assay was performed as described previously [24]. Briefly, a synthesized *SIRT1* gene was inserted in plasmid pMA-RQ and linearized with BglII (Promega, Madison, USA) to obtain 5′ mRNA fragments containing the IRE1 cleavage site (CUGCAG). To generate mutation at the cleavage site at the 5′ region of *SIRT1* mRNA, pMA-RQ-*SIRT1* was cut by ApaI/XhoI, then an ApaI-XhoI small fragment (198 bp) was removed. The mutant fragment was prepared by PCR with the mutant primer set (*SIRT1*_ApaI_m1: AGATGGGCCCTACAGGCC or *SIRT1*_ApaI_m2: AGATGGGCCCTGAAGGCC and *SIRT1*_XhoI: GGCCTCGAGCGGAGC) and ApaI/XhoI cut. The ApaI-XhoI fragment was replaced with the ApaI-XhoI mutant fragment. Mutant sequences were validated by sanger sequencing. T7 transcription kit (AM1333, Thermo Fisher Scientific, Waltham, MA, USA) was used for in vitro transcription of the 5′ region of SIRT1, and an in vitro cleavage assay of SIRT1 mRNA was performed as described earlier [25]. Finally, RNA products were incubated with or without recombinant IRE1 (E31-11G, SignalChem) at 37 °C, and fragments were separated on a 1.5% denaturing agarose gel.

### 2.12. Reverse Transcription Polymerase Chain Reaction (RT-PCR)

Total RNA from tissue was separated using TRIzol reagent (Invitrogen, Carlsbad, CA, USA). cDNA was synthesized from 1 μg of RNA with PrimeScript reverse transcript reagent Kit (Takara, Tokyo, Japan). Quantitative PCR was undertaken using TaKaRa SYBR premix Ex Taq kit (TaKaRa Bio Inc., Kusatsu, Shiga, Japan) on ABI PRISM 7500 (Applied Biosystems, Foster City, CA, USA). The comparative cycle threshold (Ct) method was used to quantify the expression levels, and each amplified product was adjusted to β-actin expression. The primer sequences of the genes were designed according to the sequence information from GenBank database (Table 1).

### 2.13. Lipid Peroxidation Measurement

Lipid peroxidation was quantified with OxiSelect^TM^ TBARS Assay Kit (STA-330, Cell Biolabs, Inc., San Diego, CA, USA) by following the protocol set by the manufacturer. Samples were allowed to react with thiobarbituric acid (TBA), and the resultant mixture was measured at 532 nm.

### 2.14. Membrane Fluidity

The formation of pyrenedecanoic acid (PDA) excimers is critical in quantifying ER membrane fluidity. Hence, a membrane fluidity kit (Axxora, Enzo Life Sciences, Farmingdale, NY, USA) was used to evaluate the formation of PDA excimers. This specific test was performed as reported earlier [25]. Briefly, ER microsomes were incubated at 25 °C with 10 μM PDA solution with PBS and 0.08% pluronic F-127. This incubation allows PDA incorporation into the membranes. Later, the microsomes were washed with PBS and resuspended in fresh PBS. Endpoint fluorescence was recorded with a SpectraMax M5 microplate reader (Molecular Devices) at the specified wavelength of 360 nm. Further, PDA excimer and monomer emissions were measured at 470 and 400 nm, respectively. To quantify relative membrane fluidity, excimer emission intensity was normalized against monomer emission.

### 2.15. Dual-Energy X-ray Absorptiometry (DXA) Scan

The percentage of fat and fat mass were measured using a cone-beam flat panel detector DXA (iNSiGHT VET DXA, Osteosys, Daegu, Korea) according to the manufacturer’s instructions. The lean body mass was deduced from the total body weight to calculate fat mass. In color-composition images, fat and lean tissue are indicated in red and green, respectively. To evaluate abdominal fat, the region of interest (ROI) was defined from the whole-body scan. The area indicates a rectangular box extending from one vertebral space to another, with the lateral border extending to the edge of the abdominal tissue. The abdominal fat percentage was calculated using the following equation: abdominal fat (DEXA)/total fat (DEXA) X100.

### 2.16. Statistical Analysis

GraphPad Prism version 8.0 (GraphPad Software, San Diego, CA, USA) was used for all statistical analyses. Two-way ANOVA followed by Tukey–Kramer post hoc test were used for multiple comparisons. All the data are presented as mean ± SEM and a *p*-value of < 0.05 was considered statistically significant.

## 3. Results

### 3.1. GABA and FCLL-GABA Regulate Body Weight Gain and Its Effect on Metabolic Profile in HFD Induced Obese Mice

GABA and FCLL-GABA products were quantified to standardize and confirm the quality of the extracted compound. As illustrated in Figure 1A,B, GABA was identified as a component indicated by the retention time (fermented *Curcuma longa* L.: 19.642 min, water extracted *Curcuma longa* L.:19.600 min). Observations show that GABA in fermented *Curcuma longa* L. is 8181.385 mg/L, whereas water extracted *Curcuma longa* L. contained 64.417 mg/L of GABA. These observations clearly show that fermented *Curcuma longa* L. is enriched with GABA. Curcumin was not detected in fermented *Curcuma longa* L. (16.742 min), but it was detected in water extracted *Curcuma longa* L. (16.758 min) (Figure 1C). GABA and FCLL-GABA supplementations prevented the body weight gain while body weight was significantly enhanced in HFD-fed mice (Figure 2A,B). Further, a Dual-energy X-ray Absorptiometry (DXA) scan indicated higher visceral adipose tissue in the HFD-fed mice than in the NCD-fed mice. On the other side, HFD fed mice with GABA and FCLL-GABA were observed to have reduced fat accumulation dose-dependently (Figure 2C). Moreover, measurement of volume from a DXA scan revealed lower orbital fat volume (%) and mass (g) in GABA and FCLL-GABA treated groups, confirming the lowered fat accumulation (Figure 2D,E). Notably, similar observations were recorded with respect to the weight of the liver, epididymal WAT (eWAT), inguinal WAT (iWAT), and brown adipose tissue (BAT) (Figure 2F–I).

### 3.2. GABA and FCLL-GABA Regulate Biochemical Characteristics and Ameliorate Hepatic Lipid Accumulation in HFD Induced Obese Mice

To evaluate the effect of GABA and FCLL-GABA on hepatic function, including hepatic lipid metabolism, we analyzed hepatic enzymes and measured lipid profiles in all the groups. As expected, HFD mice demonstrated significantly higher serum ALT and AST levels, whereas GABA and FCLL-GABA supplementation regulated these enzyme activities (Figure 3A,B). Further, lipid contents were assessed to determine the effect of GABA and FCLL-GABA on lipid homeostasis. HFD fed mice showed considerably higher serum triglyceride and total cholesterol levels than HFD mice fed with GABA and FCLL-GABA (Figure 3C–E). Moreover, hepatic triglyceride levels reveal increased hepatic lipid contents in HFD-fed mice (Figure 3F). Contrastingly, HFD-fed mice administered with GABA and FCLL-GABA showed significantly greater inhibitions in hepatic triglyceride levels than HFD fed mice. Additionally, GABA and FCLL-GABA supplementation demonstrated improved adiponectin and lowered leptin levels (Figure 3G,H).

### 3.3. GABA and Fermented Curcuma longa L. Extract Enriched with GABA Control the Levels of Adipogenesis-Related Proteins in Adipose Tissues in HFD Induced Obese Mice

Adipogenic factors were evaluated to confirm the influence of GABA and FCLL-GABA on hepatic metabolic factors. H&E staining indicated high efficiency of GABA and FCLL-GABA in reducing the size and diameter of the adipocytes (Figure 4A,B). Enhanced lipogenesis and reduced β-oxidation are clear signs of lipid accumulation [26]. Hence, lipogenesis and fatty acid oxidation markers in eWAT were analyzed to determine the molecular mechanisms of lowering eWAT mass upon GABA and FCLL-GABA administration. Concerning fatty acid biosynthesis, PPAR-γ, C/EBPα, SREBP-1c, and FAS mRNA expression levels were greater in the HFD group than in the NCD group. However, GABA and FCLL-GABA supplementation significantly lowered mRNA expression relevant to fatty acid biosynthesis (Figure 4C–F). To elucidate the contribution of reduced eWAT GABA and FCLL-GABA, adipogenic transcription factors in eWAT were evaluated. GABA and FCLL-GABA supplementation significantly reduced the protein expressions of PPAR-γ, SREBP1, C/EBPα, and FAS, more than in the HFD-fed mice (Figure 4G–K). Together, these observations suggest that GABA and FCLL-GABA regulate eWAT weight by altering adipogenesis-associated transcription factors.

### 3.4. GABA and Fermented Curcuma longa L. Extract Enriched with GABA Inhibit the Expression of Adipogenesis-Related Genes in the Liver of HFD Induced Obese Mice

The influence of GABA and FCLL-GABA on adipose tissue was examined, as abnormal WAT expansion leads to ectopic fat deposition in other tissues. To investigate, the liver from GABA and FCLL-GABA administered HFD-fed mice was evaluated, as abnormal WAT expansion leads to ectopic fat deposition in other tissues. HFD enhanced the adipogenic protein expressions more than the NCD group, while GABA and FCLL-GABA supplementation downregulated the adipogenic protein expressions (Figure 5A–D). To elucidate the contribution of reduced eWAT GABA and FCLL-GABA, adipogenic transcription factors in eWAT were evaluated. GABA and FCLL-GABA supplementation significantly reduced the protein expressions of PPAR-γ, SREBP1, C/EBPα, and FAS, more than in the HFD-fed mice (Figure 5E–I). The AMPK functions as a power switch, since it phosphorylates enzymes linked to lipid metabolism in several tissues, including the liver [27]. Additionally, previous reports indicated increased expressions of PPAR-γ, C/EBPα, and SREBP1 in the liver of HFD mice [28,29,30]. Here, hepatic p-AMPK and its linked sirtuin 1 (SIRT1) level were evaluated in HFD-fed mice with or without GABA and FCLL-GABA. The expressions of p-AMPK and SIRT1 in the liver of HFD-fed mice were lower than the NCD mice, whereas GABA and FCLL-GABA supplementation recovered the decreased expressions (Figure 5J,K).

### 3.5. GABA and Fermented Curcuma longa L. Extract Enriched with GABA Control Oxidative Stress and ER Redox Imbalance in HFD Induced Obese Mice

Accumulating evidence has demonstrated a significant association of the excess ROS production (primarily superoxide anions) with obesity and associated complications [31,32]. We first examined ROS levels in WAT of GABA and FCLL-GABA administered HFD mice. The HFD-fed mice showed higher ROS levels in the WAT than in the NCD mice, while the same was decreased upon supplementation of GABA and FCLL-GABA (Figure 6A,B). It is reported that energy metabolism-related ROS production is highly affected by the conversion of NADPH to NADP^+^, which is controlled by NADPH oxidase [33]. Thus, the NADP^+^/NADPH ratio and NADPH oxidase activity were examined in GABA and FCLL-GABA treated conditions. As expected, the NADP^+^/NADPH ratio and NADPH oxidase activity were reduced upon treatment with GABA and FCLL-GABA (Figure 6C,D). Further, Nox4 levels were significantly more increased in the HFD condition than in NCD, whereas supplementation with GABA and FCLL-GABA significantly reduced the expression of Nox4 (Figure 6E). In subcellular fraction assay, the Nox4 was confirmed to be localized in the endoplasmic reticulum (ER), verified by expressions of the ER marker protein; calnexin, mitochondria maker protein; TOMM20, cytoplasm marker protein; GAPDH and plasma membrane marker; and Na^+^/K^+^-ATPase (Figure 6F). Next, ER membrane-specific lipid peroxidation and its linked membrane fluidity were measured. HFD induced ROS production in the ER increased the lipid peroxidation, indicating oxidative degradation of lipids [25], while GABA and FCLL-GABA treatment significantly reduced the ER membrane lipid peroxidation and recovered the impaired ER membrane fluidity in the HFD model (Figure 6G,H).

### 3.6. GABA and Fermented Curcuma longa L. Extract Enriched with GABA Regulate the IRE1α Sulfonation-RIDD-SIRT1 Decay Axis and ER Stress Response in HFD Induced Obese Mice

The ER stress response is a potential therapeutic target for chronic metabolic disorders, including obesity [34]. Hence, we determined the levels of ER stress markers such as p-IRE1α, GRP78, CHOP, and sXBP-1 (Figure 7A,B). More elevated ER stress markers were observed in HFD-fed mice than in the NCD mice. Furthermore, it was observed that phosphorylation of IRE1α, subsequent XBP-1 splicing, and the expressions of GRP78 and CHOP were inhibited in GABA and FCLL-GABA treated HFD mice. It is understood that the ER stress response arms, IRE1α post-translational modifications (PTMs) such as phosphorylation, s-nitrosylation, sulfenylation, and sulfonation, play a crucial role in metabolic disorders [35]. Specifically, IRE1α sulfonation attributed to ROS production originated from an imbalance in the ratio of NADP^+^/NADPH and the associated activation of NADPH oxidase linked to energy dysmetabolism [33]. Hence, immunoprecipitation was performed with anti-sulfonate antibody to detect the oxidized form. HFD-induced obesity enhanced IRE1α sulfonation (IRE1α:SO3), but GABA and FCLL-GABA supplementation significantly inhibited the IRE1α sulfonation (Figure 7C). Next, the expressions of IRE1α-RIDD target genes such as *Blos1*, *Hgsnat*, and *Col6* were analyzed. The HFD condition significantly reduced the expression of these genes, while GABA and FCLL-GABA supplementation facilitated the recovery of these genes (Figure 7D). Further, sirtuins (SIRT) are known to exert various effects on insulin secretion, insulin sensitivity, gluconeogenesis, and glycolysis and could be a therapeutic target for a variety of metabolic diseases [36]. Among the SIRT family members, *SIRT1* is demonstrated to have a specific RIDD target sequence, “CUGCAG” [24]. Thus, to assess whether IRE1α sulfonation affects the decay of *SIRT1*, we performed an in vitro cleavage assay and observed the cleavage of *SIRT1* mRNA by IRE1α peptide in a time-dependent manner (Figure 7E,F). Mutation of GC residues in the consensus sequence abolished the cleavage of SIRT1 mRNAs by IRE1α, leading to the recovery of IRE1α-mediated decay of *SIRT1*. Moreover, *SIRT1* mRNA expression was decreased under HFD conditions and was recovered upon treatment with GABA and FCLL-GABA (Figure 7G).

### 3.7. GABA and Fermented Curcuma longa L. Extract Enriched with GABA Regulate Lipid Metabolism via AMPK-SIRT1 Signaling and Promote Browning of BAT

SIRT1 expression was examined to verify its involvement in regulating adipogenic, lipogenic, and fatty acid oxidation linked genes. Interestingly, SIRT1 was more significantly expressed in the GABA and FCLL-GABA group than in the HFD group (Figure 8A,B). Moreover, p-AMPK expression was lower in the HFD group than in the NCD group, while GABA and FCLL-GABA substantially increased p-AMPK expression (Figure 8A,B). Furthermore, treatment of HFD-fed mice with GABA and FCLL-GABA increased SIRT1 protein levels more than those observed in the HFD mice (Figure 8A,B). Further, SIRT1-mediated regulation of PGC-1α activity could play a significant role in the metabolic adaptations to energy metabolism in brown adipose tissue (BAT) [37]. Assessment observations demonstrated that PGC-1α acetylation was significantly increased under HFD conditions, whereas supplementation of GABA and FCLL-GABA resulted in decreased acetylation (Figure 8C). These observations indicate that GABA and FCLL-GABA potentially influenced the SIRT1/PGC-1α pathway. Moreover, heat map clustering analysis upon GABA and FCLL-GABA administration showed that brown fat-specific genes such as Cidea, Dio2, UCP-1, Adrb3, and PGC-1α were enriched (Figure 8D). Brown adipocytes express mitochondrial uncoupling protein 1 (UCP1), which facilitates the energy dissipation in the form of heat for thermogenesis [38]. Here, immunoblot observations reveal recovery of UCP1 expression in GABA and FCLL-GABA supplemented HFD-fed mice (Figure 8E). Consistent with these observations, immunostaining and immunofluorescence staining data indicated similar observations (Figure 8F,G). Collectively, these outcomes strongly suggest that GABA and FCLL-GABA could serve as an activator of biogenesis and thermogenic programming, thereby increasing the activity of the WAT.

## 4. Discussion

GABA is a non-protein amino acid that acts as an inhibitory neurotransmitter and is known for various pharmacological effects. This investigation demonstrated the potential beneficial effects of GABA and fermented *Curcuma longa* L. extract enriched with GABA (FCLL-GABA) on obesity and extrapolated underlying mechanisms and their positive implications on the adipose tissue of HFD-fed mice. GABA and FCLL-GABA treatment inhibited the HFD-induced NADP^+^/NADPH ratio as well as elevation of IRE1α sulfonation. Moreover, GABA and FCLL-GABA supplementation suppressed the HFD induced fat accumulation and adipogenesis and revealed that Nox4 mediates HFD-induced ROS production. Further, study observations demonstrated that upregulation of IRE1α sulfonation and associated RIDD-SIRT1 degradation could be controlled by GABA and *FCLL*-GABA treatment. Our findings suggest the involvement of the IRE1α sulfonation-SIRT1 degradation axis in the anti-obesity effects of GABA and FCLL-GABA.

Here, GABA and FCLL-GABA treatment greatly regulated the increase in body weight and fat accumulation (Figure 2) without affecting the food intake. Additionally, its supplementation reduced the adipogenesis in eWAT of HFD-induced obese mice (Figure 4). Furthermore, GABA and FCLL-GABA administration significantly downregulated adipogenic genes such as PPAR-γ and C/EBP along with their associated transcription factors such as SREBP1 and FAS. These observations signal control of the adipogenic process by GABA.

GABA and FCLL-GABA supplementation significantly inhibited ROS via regulation of NADP^+^/NADPH ratio and Nox4-mediated HFD-induced ROS accumulation (Figure 6). Enhanced NADPH oxidases increase H_2_O_2_ production and lead to increased oxidative stress with obesity [39]. NADPH oxidase-associated ROS is assumed to be the cause of energy dysmetabolism [4,5], ER dysfunction, and ROS production in the ER [6,7]. ER is a key cellular organelle that facilitates protein folding, and a chaperone-based electron transfer system is directly involved in the folding process [40,41]. Unlike the adaptive ER stress response, redox imbalance or ROS accumulation is linked to prolonged or pathological ER stress, which influences PTMs of proteins and subsequent modifications of protein function [42,43,44]. Protein dysfunction is influenced by sulfonation, which involves the oxidation of specific cysteine residues mediated by endogenous or exogenous ROS [8]. ER-derived H_2_O_2_ is produced from Nox4 and leads to ER stress [9]. Further, IRE1α sulfonation is activated by the Nox4–ER–ROS axis and is linked to cellular dysfunction [10]. The IRE1α post-translational modification-linked signaling axis, known as regulated IRE1-dependent decay (RIDD), has diverse substrates, including proteins associated with energy metabolism, by which GABA and FCLL-GABA might prevent or control obesity. SIRT1 has been identified as one of the IRE1α sulfonation-linked RIDD targets, and it also harbors a target sequence for RIDD (Figure 7E–G).

The effect of FCLL-GABA against SIRT1 decay may have important implications, considering the critical role of SIRT1 in obesity [45]. Previous investigations indicated the decreased SIRT1 expression in the eWAT in obesity [46]. Hence, SIRT1 levels were examined in the eWAT of HFD-fed mice. AMPK, a key regulator of energy metabolism, acts on eWAT and liver lipid metabolism and affects the expression of key factors involved in mitochondrial biogenesis and energy expenditure in BAT. AMPK increases NAD^+^ amounts by increasing fatty acid oxidation or possibly by enhancing its biogenesis through nicotinamide phosphoribosyltransferase (NAMPT) [47]. This increase in NAD^+^ and NAMPT enhances SIRT1 activity and induces PGC-1α deacetylation and activation [48]. AMPK activates PGC-1α through direct phosphorylation and by facilitating SIRT1-dependent PGC-1α deacetylation. In addition, it promotes oxidative metabolism in several metabolic tissues, playing a critical role in regulating the energy state of cells [49]. Study observations suggest the promotion of thermogenesis in BAT via the AMPK-SIRT1-PGC-1α pathway by GABA and FCLL-GABA (Figure 8). Collectively, the SIRT1-PGC-1α-UCP1 axis explains the GABA and FCLL-GABA induced anti-obesity mechanism. These novel observations suggest GABA and FCLL-GABA as a promising therapeutic approach in treating obese patients. Still, this investigation did not evaluate GABA-associated side effects such as upset stomach, headache, sleepiness, and muscle weakness [50]. However, GABA enriched fermented or herbal products have been reported to have no negative side effects [51]. This could be possible, as so many endogenous components in the fermented or herbal products synergize or antagonize each other where the GABA-associated side effect might be covered.

In conclusion, supplementation of GABA and FCLL-GABA reduced the risk of obesity in the HFD model by suppressing fat accumulation in the WAT. The decay of SIRT1 as a RIDD target gene is suggested to be the key mechanism involved in the attenuation of obesity conditions, along with a well-established SIRT1-PGC-1α-linked UCP1-associated energy controlling mechanism. GABA and FCLL-GABA contribute to energy homeostasis by controlling the Nox4-IRE1α sulfonation-RIDD axis. Altogether, the investigation contributed to the existing knowledge about the health benefits of GABA and FCLL-GABA and established a strong foundation for the development of FCLL-GABA as a functional ingredient in the food and pharmaceutical industry.

## Figures and Tables

**Figure 1 nutrients-14-01680-f001:**
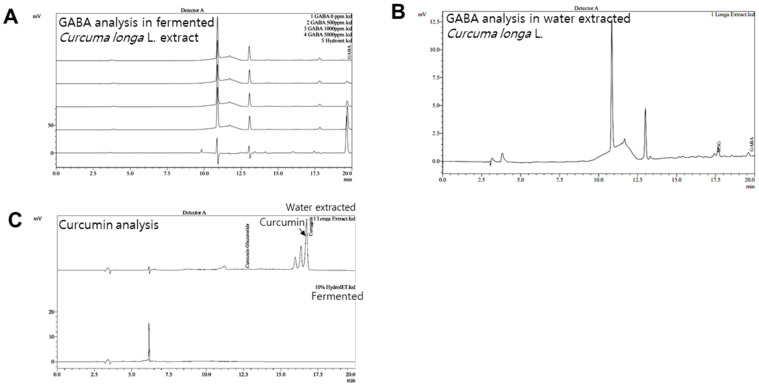
HPLC analysis of GABA in fermented and water extracted *Curcuma longa* L. extracts. Representative chromatograms of GABA standards (0, 500, 1000, and 5000 ppm) and fermented *Curcuma longa* L. extracts (**A**) and water extracted *Curcuma longa* L. (**B**). Representative chromatograms of curcumin; *Curcuma longa* L. water extract and fermented extract (**C**).

**Figure 2 nutrients-14-01680-f002:**
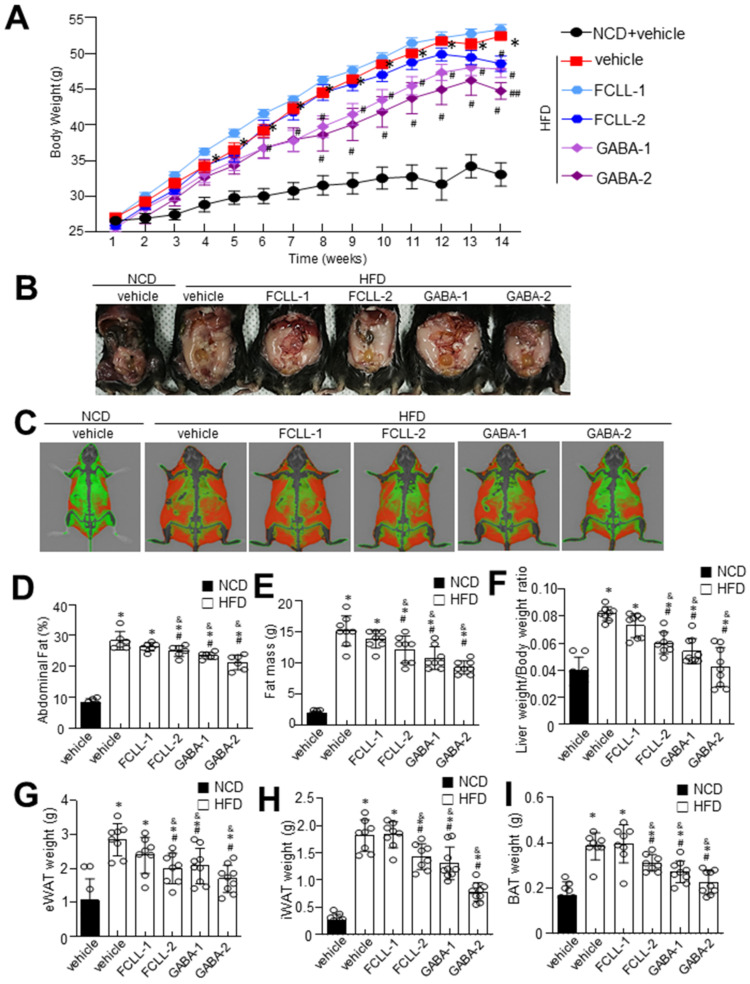
Influence of GABA and fermented *Curcuma longa* L. extract enriched with GABA on body weight and fat mass. Mice were fed with NCD or HFD with vehicle or GABA and fermented *Curcuma longa* L. extract enriched with GABA (1 or 2 g/kg) once daily for 14 weeks by oral gavage. (**A**) Changes in body weight were recorded at the indicated times. (**B**) Representative images of HFD mice supplemented with the experimental diet for 14 weeks. (**C**) Representative InAlyzer dual X-ray absorptiometry images showing fat mass. (**D**) Quantification of fat (%) and (**E**) fat mass using InAlyzer dual X-ray absorptiometry images. (**F**–**I**) Measurement of weights of liver, epididymal white adipose tissue (eWAT), inguinal white adipose tissue (iWAT), and brown adipose tissue (BAT). Data are presented as mean ± SEM (*n* = 8, * *p* < 0.05 vs. NCD + vehicle, ^&^
*p* < 0.05 vs. HFD + FCCL-1, ^#^
*p* < 0.05 vs. HFD + vehicle, ^##^
*p* < 0.05 vs. HFD + GABA-1). NCD, normal chow diet; HFD, high-fat diet; FCLL, fermented *Curcuma longa* L. extract enriched with GABA). NCD, normal chow diet; HFD, high-fat diet; FCLL, fermented *Curcuma longa* L. extract enriched with GABA; FCLL-1 and FCLL-2, HFD fed mice received 1 and 2 g/kg fermented *Curcuma Longa* L. enriched with GABA (FCLL-GABA), respectively; GABA-1 and GABA-2, HFD fed mice receiving 1 and 2 g/kg GABA, respectively.

**Figure 3 nutrients-14-01680-f003:**
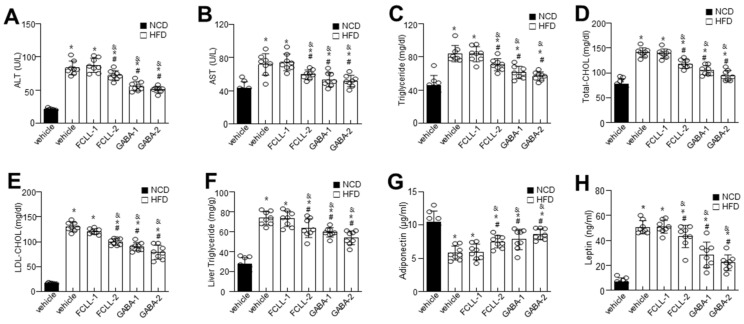
Influence of GABA and fermented *Curcuma longa* L. extract enriched with GABA on serum biochemicals profile and lipid accumulation. Mice were fed with NCD or HFD with vehicle or GABA and fermented *Curcuma longa* L. extract enriched with GABA (1 or 2 g/kg) once daily for 14 weeks by oral gavage. Biochemical analyses of serum samples. Levels of ALT (**A**), AST (**B**), Triglyceride (**C**), Total cholesterol (**D**), LDL-cholesterol (**E**), liver triglyceride (**F**), adiponectin (**G**), and leptin (**H**). Data are presented as mean ± SEM (*n* = 8, * *p* < 0.05 vs. NCD + vehicle, ^&^
*p* < 0.05 vs. HFD + FCCL-1, ^#^
*p* < 0.05 vs. HFD + vehicle). NCD, normal chow diet; HFD, high-fat diet; FCLL-1 and FCLL-2, HFD fed mice received 1 and 2 g/kg fermented *Curcuma Longa* L. enriched with GABA (FCLL-GABA), respectively; GABA-1 and GABA-2, HFD fed mice receiving 1 and 2 g/kg GABA, respectively; ALT, alanine aminotransaminase; AST, aspartate transaminase; LDL, low-density lipoproteins.

**Figure 4 nutrients-14-01680-f004:**
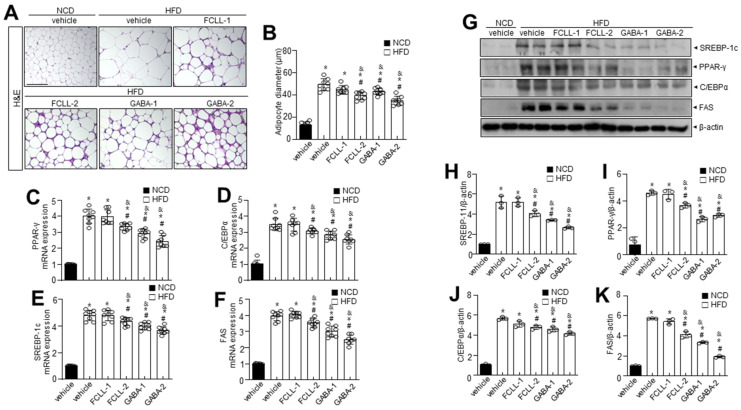
GABA and fermented *Curcuma longa* L. extract enriched with GABA determine adipose tissue expansion and adipogenic factors in eWAT. Mice were fed with NCD or HFD with vehicle or GABA and GABA-enriched *Curcuma longa* L. extract (1 or 2 g/kg) once daily for 14 weeks by oral gavage. (**A**) H&E staining in eWAT. Scale bar = 50 µm. (**B**) The average diameter of adipocytes in eWAT. (**C**–**F**) PPAR-γ, C/EBPα, SREBP-1c, and FAS were measured in eWAT by qRT-PCR. (**G**) Immunoblotting of SREBP-1c, PPAR-γ, C/EBPα, FAS, and β-actin expressions in eWAT and (**H**–**K**) respective quantitative analysis of protein expressions. Data are presented as mean ± SEM (*n* = 8, * *p* < 0.05 vs. NCD + vehicle, ^&^
*p* < 0.05 vs. HFD + FCCL-1, ^#^
*p* < 0.05 vs. HFD + vehicle). eWAT, epididymal white adipose tissue; NCD, normal chow diet; HFD, high-fat diet; FCLL-1 and FCLL-2, HFD fed mice received 1 and 2 g/kg fermented *Curcuma Longa* L. enriched with GABA (FCLL-GABA), respectively; GABA-1 and GABA-2, HFD fed mice receiving 1 and 2 g/kg GABA, respectively.

**Figure 5 nutrients-14-01680-f005:**
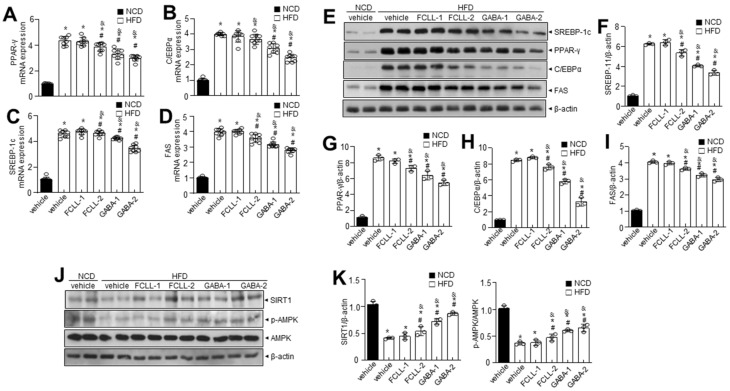
GABA and fermented *Curcuma longa* L. extract enriched with GABA regulates adipogenesis-related genes in the liver of HFD induced obese mice. Mice were fed with NCD or HFD with vehicle or GABA and GABA-enriched *Curcuma longa* L. extract (1 or 2 g/kg) once daily for 14 weeks by oral gavage. (**A**–**D**) PPAR-γ, C/EBPα, SREBP-1c, and FAS were measured in liver tissue by qRT-PCR. (**E**) Immunoblotting of SREBP-1c, PPAR-γ, C/EBPα, FAS, and β-actin expressions in liver tissue and (**F**–**I**) respective quantitative analysis of protein expressions. (**J**) Immunoblotting of SIRT-1, p-AMPK, AMPK, and β-actin expressions was performed in liver tissue and (**K**) respective quantitative analysis of protein expressions. Data are presented as mean ± SEM (*n* = 8, * *p* < 0.05 vs. NCD + vehicle, ^&^
*p* < 0.05 vs. HFD + FCCL-1, ^#^
*p* < 0.05 vs. HFD + vehicle). NCD, normal chow diet; HFD, high-fat diet; FCLL-1 and FCLL-2, HFD fed mice received 1 and 2 g/kg fermented *Curcuma Longa* L. enriched with GABA (FCLL-GABA), respectively; GABA-1 and GABA-2, HFD fed mice receiving 1 and 2 g/kg GABA, respectively.

**Figure 6 nutrients-14-01680-f006:**
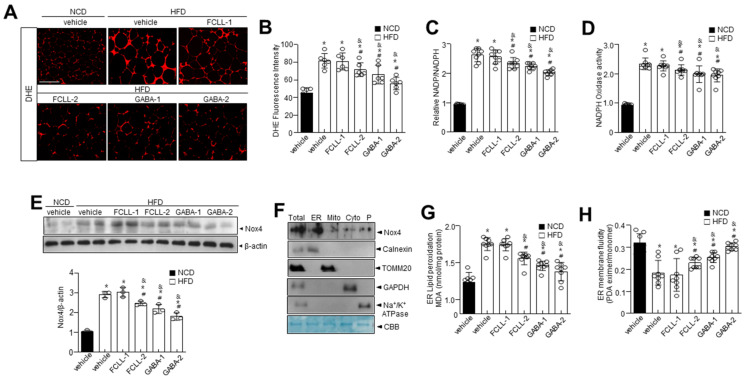
GABA and fermented *Curcuma longa* L. extract enriched with GABA regulate oxidative stress and ER redox imbalance in HFD model. Mice received vehicle or GABA and GABA-enriched *Curcuma longa* L. extract (1 or 2 g/kg) once daily for 14 weeks by oral gavage. Representative dihydroethidium (DHE)-stained images depicting ROS production (**A**) and quantification (**B**) in each condition. Scale bars = 100 µm. eWAT lysates were analyzed for NADP/NADPH ratio (**C**) and NADPH oxidase activity (**D**). (**E**) Expression of Nox4 and β-actin in eWAT and respective quantitative analysis of protein expressions. (**F**) Subcellular fractions of lysates were analyzed by immunoblotting. (**G**) eWAT lysates were analyzed for lipid peroxidation levels. (**H**) Analysis of membrane fluidity in the ER fraction. Data are presented as mean ± SEM (*n* = 8, * *p* < 0.05 vs. NCD + vehicle, ^&^
*p* < 0.05 vs. HFD + FCCL-1, ^#^
*p* < 0.05 vs. HFD + vehicle). NCD, normal chow diet; HFD, high-fat diet; FCLL-1 and FCLL-2, HFD fed mice received 1 and 2 g/kg fermented *Curcuma Longa* L. enriched with GABA (FCLL-GABA), respectively; GABA-1 and GABA-2, HFD fed mice receiving 1 and 2 g/kg GABA, respectively.

**Figure 7 nutrients-14-01680-f007:**
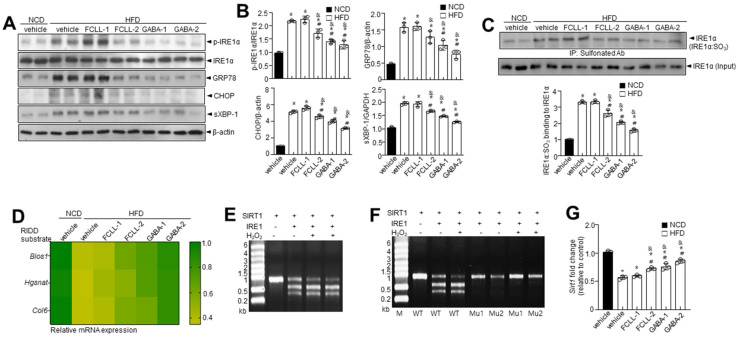
GABA and fermented *Curcuma longa* L. extract enriched with GABA diminish ER stress and IRE1α sulfonation of axis in HFD model. Mice received vehicle or GABA and GABA-enriched *Curcuma longa* L. extract (1 or 2 g/kg) once daily for 14 weeks by oral gavage. (**A**) Immunoblotting of p-IRE1α, IRE1α, GRP78, CHOP, sXBP-1, and β-actin expressions in eWAT. (**B**) Quantitative analysis of protein expressions. (**C**) Sulfonation of IRE1α in eWAT tissue. (**D**) Heatmap depicting mRNA expression of the genes identified as RIDD substrates in eWAT. In vitro cleavage assay at 1.5% denaturing agarose gel electrophoresis of *SIRT1* substrate cleaved by IRE1α in the presence or absence of 100 μM H_2_O_2_ (**E**) and with its two mutant mRNAs (**F**). (**G**) The mRNA level of SIRT1. Data are presented as mean ± SEM (*n* = 8, * *p* < 0.05 vs. NCD + vehicle, ^&^
*p* < 0.05 vs. HFD + FCCL-1, ^#^
*p* < 0.05 vs. HFD + vehicle). eWAT, epididymal white adipose tissue; NCD, normal chow diet; HFD, high-fat diet; FCLL-1 and FCLL-2, HFD fed mice received 1 and 2 g/kg fermented *Curcuma Longa* L. enriched with GABA (FCLL-GABA), respectively; GABA-1 and GABA-2, HFD fed mice receiving 1 and 2 g/kg GABA were assigned, respectively.

**Figure 8 nutrients-14-01680-f008:**
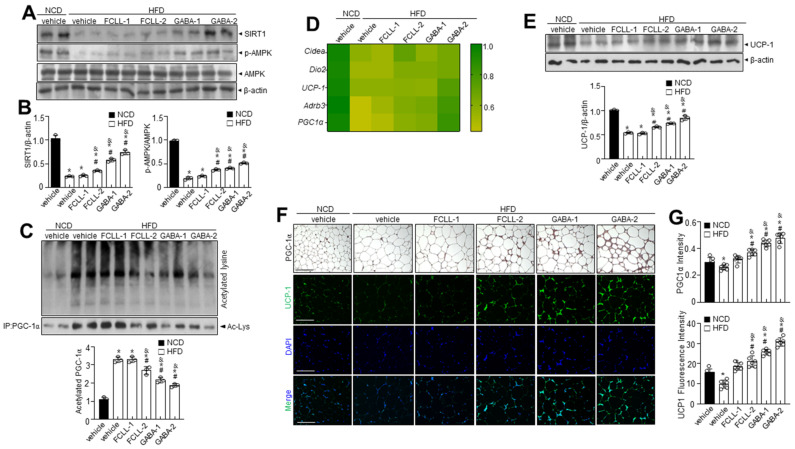
GABA and fermented *Curcuma longa* L. extract enriched with GABA promote thermogenic program in brown adipose tissue. Mice received vehicle or GABA and fermented *Curcuma longa* L. extract enriched with GABA (1 or 2 g/kg) once daily for 14 weeks by oral gavage. (**A**) Immunoblotting of SIRT1, p-AMPK, AMPK, and β-actin in BAT and (**B**) respective quantitative analysis of protein expression. (**C**) Immunoprecipitation using an antibody against PGC-1α to evaluate acetylation levels. (**D**) Heatmap depicting mRNA expression of brown fat-specific genes. (**E**) Immunoblotting of UCP1 and β-actin in the BAT and quantitative analysis of protein expressions. (**F**,**G**) Representative images of the PGC-1α immunohistochemical staining or the immunofluorescence staining of UCP1 on BAT sections and respective quantification of fluorescence intensity. Data are presented as mean ± SEM (*n* = 8, * *p* < 0.05 vs. NCD + vehicle, ^&^
*p* < 0.05 vs. HFD + FCCL-1, ^#^
*p* < 0.05 vs. HFD + vehicle). BAT, brown adipose tissue, NCD, normal chow diet; HFD, high-fat diet; FCLL-1 and FCLL-2, HFD fed mice received 1 and 2 g/kg fermented *Curcuma Longa* L. enriched with GABA (FCLL-GABA), respectively; GABA-1 and GABA-2, HFD fed mice receiving 1 and 2 g/kg GABA were assigned, respectively.

**Table 1 nutrients-14-01680-t001:** Primer sequences used for RT-PCR.

Gene		Primer Sequences (5′–3′)
SREBP-1c (NC_000077.7)	Forward Reverse	CTGTTGGTGCTCGTCTCCT TTGCGATGCCTCCAGAAGTA
PPARγ (NC_000072.7)	Forward Reverse	GATGACAGCGACTTGGCAAT TGTAGCAGGTTGTCTTGAATGT
FAS (NC_000085.7)	Forward Reverse	ATCCTGGCTGACGAAGACTC TGCTGCTGAGGTTGGAGAG
CEBPα (NC_000073.7)	Forward Reverse	GACTTGGTGCGTCTAAGATGAG TAGGCATTGGAGCGGTGAG
Cidea (NC_000084.7)	Forward Reverse	GCCTGCAGGAACTTATCAGC GCCTGCAGGAACTTATCAGC
Dio2 (NC_000078.7)	Forward Reverse	CTGCGCTGTGTCTGGAAC GGAATTGGGAGCATCTTCAC
UCP-1 (NC_000074.7)	Forward Reverse	GGCCTCTACGACTCAGTCCA TAAGCCGGCTGAGATCTTGT
Adrb3 (NC_000074.7 )	Forward Reverse	ACTGCTAGCATCGAGACCTTG AGGGTTGGTGACAGCTAGG
PGC-1α (NC_000071.7)	Forward Reverse	GAAAGGGCCAAACAGAGAGA GTAAATCACACGGCGCTCTT
β-actin (NC_000071.7)	Forward Reverse	AAGACCTCTATGCCAACACAGT AGCCAGAGCAGTAATCTCCTTC

## Data Availability

The data presented in this study are available upon request from the corresponding author.

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
