# Peer review of "GABA and Fermented Curcuma longa L. Extract Enriched with GABA Ameliorate Obesity through Nox4-IRE1α Sulfonation-RIDD-SIRT1 Decay Axis in High-Fat Diet-Induced Obese Mice"

_nutrients, 2022, doi:10.3390/nu14081680_

Round 1
Reviewer 1 Report
The paper entitled "GABA and fermented Curcuma longa L. extract enriched with GABA ameliorate obesity through Nox4-IRE1α sulfonation-RIDD-SIRT1 decay axis in high-fat diet-induced obese mice" is a robust work.
I have some criticism:
- the statistical analysis is incorrect. If you analyze in the same figure both dietary regimens and supplements the correct analysis is the two way ANOVA.
- In figure 1 A why GABA-1 e GABA-2 mice are so different and in figure 1 B the weights are similar?
- Why did the authors choose 14 weeks of treatment?
- the authors did not evaluate the behavioural scores of animals. This is a lack of crucial information to establish the absence of GABAergic side effects.
minor point
the figures should be modified in the box and jitter plots to show the data.
Author Response
Q1. the statistical analysis is incorrect. If you analyze in the same figure both dietary regimens and supplements, the correct analysis is the two way ANOVA.
A1. Following the comment, we analyzed the data using the two-way ANOVA.
Q2. In figure 1 A why GABA-1 e GABA-2 mice are so different and in figure 1 B the weights are similar
A2. We appreciate the reviewer's valuable comment. Following the comment, we have performed the statistical analysis between the group. Administration of GABA-2 (relatively higher dose) significantly reduced the bodyweight than the GABA-1. We have attached raw observations on the bodyweight for your reference. RAW DATA for body weight changes
Bar graph to show the GABA-1 and -2 groups
Q3. Why did the authors choose 14 weeks of treatment?
A3. Generally, researchers adopt a 12-week regime for HFD models, but an extended period of up to 24 weeks is adopted to investigate metabolism-linked mechanisms. However, this investigation was extended by 2 weeks to clearly observe the expression of Nox4 and the associated IRE-1 sulfonation axis. Also, study observations showed that ER stress response associated expressions are similar to 12, 14, 16, and 24-week HFD models.
Q4. the authors did not evaluate the behavioural scores of animals. This is a lack of crucial information to establish the absence of GABAergic side effects.
A4. We appreciate the valuable comment. GABA is associated with side effects such as upset stomach, headache, sleepiness, and muscle weakness. However, there are no standard methods to evaluate all these side effects at once. Thus, in this investigation, side effects were not evaluated. In the revised manuscript, we have added this as a study limitation. Updated text in the discussion section is as follows,
“GABA-associated side effects such as upset stomach, headache, sleepiness, and muscle weakness [50] were not evaluated in this investigation. However, there are reports that GABA-enriched fermented or herbal products do not show side effects [51]. This could be possible as so many endogenous components in the fermented or herbal products synergize or antagonize each other where the GABA-associated side effect might be covered”.
minor point
Q1. the figures should be modified in the box and jitter plots to show the data.
A1. As suggested, we have modified figures as follows.

Reviewer 2 Report
Paper by Lee et al documents the anti-obesity potential of GABA neurotransmitter and Curcuma longa L. extract supplemented with GABA in a high fat diet mouse model.
Despite the scientific soundness by which the Authors approached this study, it still seems necessary to address some relevant points to improve the manuscript.
Therefore:
1) The abstract and the introduction should be more fluent, being made by short sentences or frequent “further” or “additionally”, which make the text difficult to read.
2) In view of the large number of results obtained, the discussion is worthy of further implementation.
3) The description of the HPLC analysis of GABA in fermented and water-extracted Curcuma longa L. extracts is lacking in “Materials and Methods” section. What other components does the extract contain? Authors must specify this. In addition, disclose how the extract is administered to mice.
4) In paragraph 2.11. of "Material and Methods" section, please, list in a table the oligonucleotide primer sequences used for the quantitative real-time PCR analysis and their GenBank accession number.
5) Please, clarify the title of paragraph 2.10. of “Material and Methods” section and the titles of the captions in Figures 4 and 5, as well as make the sentence of lines 365-368 more understandable by commenting about Figures 7E and 7F and checking their legend. Similarly, please, explain and comment sufficiently results of Figure 6F. In addition, the results of Figure 5J, concerning the enzyme SIRT1, are not mentioned in the text. Therefore, please, make a comment on this result as well.
6)Please, report a quantitative analysis of protein expressions of Figures 7C and 8C.
7) Please correct some mistake like this: the sentence of lines 284-285 appears twice, or, please, change “0.5” to “0.05” in line 189. Then, please, correctly place the stars of the statistics in Figure 2B, since they appear to refer to the FCLL-1 group rather than the vehicle group or still, please, correct “S-nitrosylation” to “sulfonation” in the caption of Figure 7C. These are just examples; thus, Authors must check thoroughly the text.
Author Response
Rebuttal Letter
Reviewer 2
Paper by Lee et al documents the anti-obesity potential of GABA neurotransmitter and Curcuma longa L. extract supplemented with GABA in a high fat diet mouse model.
Despite the scientific soundness by which the Authors approached this study, it still seems necessary to address some relevant points to improve the manuscript.
Therefore:
1) The abstract and the introduction should be more fluent, being made by short sentences or frequent “further” or “additionally”, which make the text difficult to read.
A1. As suggested, we have updated the abstract and the introduction parts to improve the readability. The revised abstract and introduction parts are as follows,
“Abstract
Gamma-aminobutyric acid (GABA) is a natural amino acid with antioxidant activity and is often considered to have therapeutic potential against obesity. Obesity has long been linked to ROS and ER stress, but the effect of GABA on ROS-associated ER stress axis has not been thoroughly explored. Thus, in this study, GABA and fermented Curcuma longa L. extract enriched with GABA (FCLL-GABA) on ROS-related ER stress axis, Inositol-requiring transmembrane kinase/endoribonuclease 1α (IRE1α) sulfonation was examined with HFD model to determine the underlying anti-obesity mechanism. Here, GABA and FCLL-GABA supplementations significantly inhibited the weight gain in HFD fed mice. The GABA and FCLL-GABA supplementation lowered the expressions of adipogenic transcription factors like PPAR-γ, C/EBPα, FAS, and SREBP-1c in white adipose tissue (WAT) and liver from HFD-fed mice. The enhanced hyper-nutrient dysmetabolism-based NADPH oxidase (Nox) 4 and the resultant IRE1α sulfonation-RIDD-SIRT1 decay under HFD conditions were controlled with GABA and FCLL-GABA. Notably, GABA and FCLL-GABA administration significantly increased AMPK and sirtuin 1 (SIRT1) levels in WAT of HFD-fed mice. These significant observations indicate that ER-localized Nox4-induced IRE1α sulfonation results in the decay of SIRT1 as a novel mechanism behind the positive implications of GABA on obesity. Moreover, the investigation lays a firm foundation for the development of FCLL-GABA as a functional ingredient.”
“Introduction
Obesity is a growing global health concern associated with multiple chronic diseases, like type 2 diabetes, multiple cancers, cardiovascular diseases, and hypertension [1]. White adipose tissue (WAT) plays a crucial role in regulating systemic energy and is essential for the proper functioning of several metabolic tasks to manage obesity-associated chronic diseases. However, it loses its functional capabilities with developing obesity as it impairs WAT abilities to store surplus energy. This leads to ectopic fat accumulation in various tissues that influence metabolic homeostasis [2]. Hence, therapeutic initiatives targeting lipogenesis and lipid dysmetabolism in obesity and linked clinical conditions have potential therapeutic applications [3]. During WAT-associated metabolic processes, the NADPH oxidase and its linked superoxide need to be tightly controlled for metabolic homeostasis. An increase in the activity of NADPH oxidase enhances the production of superoxides leading to metabolic disorders due to increased oxidative stress. NADPH oxidase-associated reactive oxygen species (ROS) are assumed to be the cause of the reduced insulin responsiveness [4,5], endoplasmic reticulum (ER) dysfunction, and ROS production in the ER [6,7]. During stress, protein dysfunction is influenced by sulfonation, which involves the oxidation of specific cysteine residues mediated by endogenous or exogenous ROS [8]. Moreover, ER-derived H2O2 is produced from Nox4 and leads to ER stress [9]. Inositol-requiring enzyme 1α (IRE1α) sulfonation is activated via NADPH oxidase 4 (Nox4)-ER-ROS axis and is linked to cellular dysfunction [10]. The IRE1α post-translational modification (PTM)-linked signaling axis, known as regulated IRE1-dependent decay (RIDD), has diverse substrates, including proteins associated with glucose metabolism. Besides, Sirtuin (SIRT) family genes played critical roles in metabolic disorders. Specifically, ER stress axis-linked SIRT is a frequently investigated pathway, but the application of the IRE-1-RIDD axis to SIRT fate is yet to be defined.
Gamma-aminobutyric acid (GABA) is an amino acid that serves as an inhibitory neurotransmitter for the central nervous system (CNS). Several reports identified GABA as an anti-obesity compound that regulates obesity through anti-inflammatory, antioxidant, and properties that improve glucose metabolism [11-15]. In a high-fat diet (HFD) model, supplementation of GABA suppressed weight gain and reduced fat mass [12,14,15]. Similarly, administration of GABA in lean and diabetic mice revealed positive implications on glucose metabolism with an increase in β-cell mass and function without affecting the bodyweight [13,16,17]. Moreover, clinical trials suggest that GABA potentially enhances circulating insulin and glucagon [18,19]. However, GABA therapeutic applications were restricted by potential side effects, including upset stomach, headache, sleepiness, and muscle weakness. Hence, GABA-enriched natural products such as fermented food or herbal medicine have been proposed as a promising alternative strategy to treat metabolic disorders. For ages, fermentation has been more popular and relatively simple than chemical/biotechnological processes. Thus, GABA-enriched fermented foods might be the best choice to replace GABA. Further, accumulated investigations with Curcuma longa L. on obesity prompted us to evaluate the effects of fermented Curcuma longa L. extract enriched with GABA (FCLL-GABA).
In this study, the effects of GABA and fermented Curcuma longa L. extract enriched with GABA on obesity were evaluated using the HFD model and made an effort to extrapolate the mechanism responsible for its anti-obesity effect.
2) In view of the large number of results obtained, the discussion is worthy of further implementation.
A2. Following the reviewer’s comment, we have updated the discussion sections as follows.
“GABA is a non-protein amino acid that acts as an inhibitory neurotransmitter and is known for various pharmacological effects. This investigation demonstrated the potential beneficial effects of GABA and fermented Curcuma longa L. extract enriched with GABA (FCLL-GABA) on obesity and extrapolated underlying mechanisms and their positive implications on the adipose tissue of HFD-fed mice. GABA and FCLL-GABA treatment inhibited HFD-induced NADP+/NADPH ratio as well as elevation of IRE1α sulfonation. Moreover, GABA and FCLL-GABA supplementation suppressed the HFD induced fat accumulation and adipogenesis and revealed that Nox4 mediates HFD-induced ROS production. Further, study observations demonstrated that upregulation of IRE1α sulfonation and associated RIDD-SIRT1 degradation could be controlled by GABA and FCLL-GABA treatment. Our findings suggest the involvement of the IRE1α sulfonation-SIRT1 degradation axis in the anti-obesity effects of GABA and FCLL-GABA. Here, GABA and FCLL-GABA treatment greatly regulated the increase in body weight and fat accumulation (Figure 2) without affecting the food intake. Additionally, its supplementation reduced the adipogenesis in eWAT of HFD-induced obese mice (Figure 4). Also, GABA and FCLL-GABA administration significantly downregulated adipogenic genes such as PPAR-γ and C/EBP along with their associated transcription factors like SREBP1 and FAS. These observations signal control of the adipogenic process by GABA.
GABA and FCLL-GABA supplementation significantly inhibited ROS via regulation of NADP+/NADPH ratio and Nox4 mediated HFD-induced ROS accumulation (Figure 6). Enhanced NADPH oxidases increase H2O2 production and lead to increased oxidative stress with obesity [20]. NADPH oxidase-associated ROS are assumed to be the cause of energy dysmetabolism [4,5], ER dysfunction, and ROS production in the ER [6,7]. ER is a key cellular organelle that facilitates protein folding, and a chaperone-based electron transfer system is directly involved in the folding process [21,22]. Unlike the adaptive ER stress response, redox imbalance or ROS accumulation is linked to prolonged or pathological ER stress, which influences PTMs of proteins and subsequent modifications of protein function [23-25]. Protein dysfunction is influenced by sulfonation, which involves the oxidation of specific cysteine residues mediated by endogenous or exogenous ROS [8]. ER-derived H2O2 is produced from Nox4 and leads to ER stress [9]. Further, IRE1α sulfonation is activated by the Nox4-ER-ROS axis and is linked to cellular dysfunction [10]. The IRE1α post-translational modification-linked signaling axis, known as regulated IRE1-dependent decay (RIDD), has diverse substrates, including proteins associated with energy metabolism, which GABA and FCLL-GABA might prevent or control obesity. SIRT1 has been identified as one of the IRE1α sulfonation-linked RIDD targets, and it also harbors a target sequence for RIDD (Figure 7E-G).
The effect of FCLL-GABA against SIRT1 decay may have important implications, considering the critical role of SIRT1 in obesity [26]. Previous investigations indicated the decreased SIRT1 expression in the eWAT in obesity [27]. Hence, SIRT1 levels were examined in the eWAT of HFD-fed mice. AMPK, a key regulator of energy metabolism, acts on eWAT and liver lipid metabolism and affects the expression of key factors involved in mitochondrial biogenesis and energy expenditure in BAT. AMPK increases NAD+ amounts by increasing fatty acid oxidation or possibly by enhancing its biogenesis through nicotinamide phosphoribosyltransferase (NAMPT) [28]. This increase in NAD+ and NAMPT enhances SIRT1 activity and induces PGC-1α deacetylation and activation [29]. AMPK activates PGC-1α through direct phosphorylation and by facilitating SIRT1-dependent PGC-1α deacetylation. Also, it promotes oxidative metabolism in several metabolic tissues, playing a critical role in regulating the energy state of cells [30]. Study observations suggest the promotion of thermogenesis in BAT via the AMPK-SIRT1-PGC-1α pathway by GABA and FCLL-GABA (Figure 8). Collectively, the SIRT1-PGC-1α-UCP1 axis explains the GABA and FCLL-GABA induced anti-obesity mechanism. These novel observations suggest GABA and FCLL-GABA as a promising therapeutic approach in treating obese patients. Still, this investigation did not evaluate GABA-associated side effects such as upset stomach, headache, sleepiness, and muscle weakness [31]. However, GABA enriched fermented or herbal products have been reported to have no negative side effects [32]. This could be possible as so many endogenous components in the fermented or herbal products synergize or antagonize each other where the GABA-associated side effect might be covered.
In conclusion, supplementation of GABA and FCLL-GABA reduced the risk of obesity in the HFD model by suppressing fat accumulation in the WAT. The decay of SIRT1 as a RIDD target gene is suggested to be the key mechanism involved in the attenuation of obesity conditions along with a well-established SIRT1-PGC-1α-linked UCP1-associated energy controlling mechanism. GABA and FCLL-GABA contribute to energy homeostasis by controlling the Nox4-IRE1α sulfonation-RIDD axis. Altogether, the investigation contributed to the existing knowledge about the health benefits of GABA and FCLL-GABA and established a strong foundation for the development of FCLL-GABA as a functional ingredient in the food and pharmaceutical industry.”
3) The description of the HPLC analysis of GABA in fermented and water-extracted Curcuma longa L. extracts is lacking in “Materials and Methods” section. What other components does the extract contain? Authors must specify this. In addition, disclose how the extract is administered to mice.
Q3-1. The description of the HPLC analysis of GABA in fermented and water-extracted Curcuma longa L. extracts is lacking in “Materials and Methods” section.
A3-1. Following the reviewer’s comment, we have updated “Material and Methods” section with HPLC analysis as follows,
“2.2. High Performance Liquid Chromatography (HPLC) Analysis
The Elite Lachrom HPLC-DAD system with UV (DAD) detector (Hitachi High-Technologies Co., Tokyo, Japan) and CAPCELL PAK C18 (UG120, 4.6*250 mm 5 μm) column was used to perform HPLC analysis. The column temperature was 40 °C, and the injection volume was 10 μL. The mobile phases were composed of solvents acetonitrile (CAN), 10mM Phosphoric buffer (pH2.4), and methanol. The run time was 25 min, and the mobile phase process gradient flow was as follows: (A)/(B)/(C) = 1.5/97/1.5 (0 min) → (A)/(B)/(C) = 45//10/45 (0–25 min). The mobile phase flow rate was 1.0 ml/min, and the wavelength of the UV (DAD) detector was set at 280 nm”.
Q3-2. What other components does the extract contain?
A3-2. We appreciate the valuable comment. Our investigation concentrated on evaluating the effect of GABA on obesity. Thus, only GABA and fermented Curcuma long L. extract enriched with GABA (FCLL-GABA) were targeted. However, we confirmed the absence of curcumin in our test products as it may influence anti-obesity effects.
Q3-3. In addition, disclose how the extract is administered to mice.
A3-3. GABA, FCLL-GABA and water (vehicle) were administered daily via oral gavage. Following the reviewer’s comment, we have updated the information in the revised manuscript.
4) In paragraph 2.11. of "Material and Methods" section, please, list in a table the oligonucleotide primer sequences used for the quantitative real-time PCR analysis and their GenBank accession number.
A4. Following the reviewer’s comment, we have updated the Material and Methods sections as follows.
Table 1. Primer sequences used for RT-PCR.
Gene |
|
Primer Sequences (5′–3′) |
SREBP-1c (NC_000077.7) |
Forward Reverse |
CTGTTGGTGCTCGTCTCCT TTGCGATGCCTCCAGAAGTA |
PPARγ (NC_000072.7) |
Forward Reverse |
GATGACAGCGACTTGGCAAT TGTAGCAGGTTGTCTTGAATGT |
FAS (NC_000085.7) |
Forward Reverse |
ATCCTGGCTGACGAAGACTC TGCTGCTGAGGTTGGAGAG |
CEBPα (NC_000073.7) |
Forward Reverse |
GACTTGGTGCGTCTAAGATGAG TAGGCATTGGAGCGGTGAG |
Cidea (NC_000084.7) |
Forward Reverse |
GCCTGCAGGAACTTATCAGC GCCTGCAGGAACTTATCAGC |
Dio2 (NC_000078.7) |
Forward Reverse |
CTGCGCTGTGTCTGGAAC GGAATTGGGAGCATCTTCAC |
UCP-1 (NC_000074.7) |
Forward Reverse |
GGCCTCTACGACTCAGTCCA TAAGCCGGCTGAGATCTTGT |
Adrb3 ( NC_000074.7 ) |
Forward Reverse |
ACTGCTAGCATCGAGACCTTG AGGGTTGGTGACAGCTAGG |
PGC-1α (NC_000071.7) |
Forward Reverse |
GAAAGGGCCAAACAGAGAGA GTAAATCACACGGCGCTCTT |
β-actin (NC_000071.7) |
Forward Reverse |
AAGACCTCTATGCCAACACAGT AGCCAGAGCAGTAATCTCCTTC |
5) Please, clarify the title of paragraph 2.10. of “Material and Methods” section and the titles of the captions in Figures 4 and 5, as well as make the sentence of lines 365-368 more understandable by commenting about Figures 7E and 7F and checking their legend.
Similarly, please, explain and comment sufficiently results of Figure 6F. In addition, the results of Figure 5J, concerning the enzyme SIRT1, are not mentioned in the text. Therefore, please, make a comment on this result as well.
Q5-1. Please, clarify the title of paragraph 2.10. of “Material and Methods” section
A5-1. Following the reviewer’s comment, we have updated 2.10. of “Material and Methods” section as follows,
2.10. In vitro IRE1α-mediated SIRT1 cleavage assay
Q5-2: the titles of the captions in Figures 4 and 5,
A5-2: As suggested, captions were updated in Figures 4 and 5. Figure 4 describes the effect of GABA on adipose tissue and adipogenic factors, and figure 5 describes the effect of GABA on adipogenesis-related genes in the liver. These are the key parameters before describing ROS and ER stress-related mechanisms beginning from figure 6. We hope that revisions on captions are more precise and understandable.
Q5-3: make the sentence of lines 365-368 more understandable by commenting about Figures 7E and 7F
A5-3: As suggested, we have modified the relevant sentences. Revised sentences as follows,
“Thus, to assess whether IRE1α sulfonation affects the decay of SIRT1, we performed an in vitro cleavage assay and observed the cleavage of SIRT1 mRNA by IRE1α peptide, in a time-dependent manner (Figure 7E-F). Mutation of GC residues in the consensus sequence abolished the cleavage of SIRT1 mRNAs by IRE1α, leading to the recovery of IRE1α-mediated decay of SIRT1”.
Q5-4: and checking their legend.
A5-4: Following the comment, we have updated the mentioned part as follows.
“Figure 7. GABA and fermented Curcuma longa L. extract enriched with GABA diminish ER stress and IRE1α sulfonation of axis in HFD model. Mice received vehicle or GABA and GABA-enriched Curcuma longa L. extract (1 or 2 g/kg) once daily for 14 weeks by oral gavage. (A) Immunoblotting of p-IRE1α, IRE1α, GRP78, CHOP, sXBP-1, and β-actin expressions in eWAT. (B) Quantitative analysis of protein expressions. (C) Sulfonation of IRE1α in eWAT tissue. (D) Heatmap depicting mRNA expression of the genes identified as RIDD substrates in eWAT. In vitro cleavage assay at 1.5% denaturing agarose gel electrophoresis of SIRT1 substrate cleaved by IRE1α in the presence or absence of 100 μM H2O2 (E) and with its two mutant mRNAs (F). (G) The mRNA level of SIRT1. Data are presented as mean ± SEM (n = 8, *p < 0.05 vs NCD + vehicle, &p < 0.05 vs HFD + FCCL-1, #p < 0.05 vs HFD + vehicle). eWAT; epididymal white adipose tissue, NCD, normal chow diet; HFD, high-fat diet; FCLL-1 and FCLL-2, HFD fed mice received 1 and 2 g/kg fermented Curcuma Longa L. enriched with GABA (FCLL-GABA), respectively; GABA-1 and GABA-2, HFD fed mice receiving 1 and 2 g/kg GABA were assigned, respectively.”
Q5-5: Similarly, please, explain and comment sufficiently results of Figure 6F
A5-5: Following the comment, we have commented on the observations described in figure 6F. Revised text as follows,
“In subcellular fraction assay, the Nox4 was confirmed to be localized in the endoplasmic reticulum (ER), verified by expressions of the ER marker protein; calnexin, mitochondria maker protein; TOMM20, cytoplasm marker protein; GAPDH and plasma membrane marker; Na+/K+-ATPase (Figure 6F). Next, ER membrane-specific lipid peroxidation and its linked membrane fluidity were measured. HFD induced ROS production in ER increased the lipid peroxidation indicating oxidative degradation of lipids [33], while GABA and FCLL-GABA treatment significantly reduced the ER membrane lipid peroxidation and recovered the impaired ER membrane fluidity in the HFD model (Figure 6G-H).”
Q5-6: In addition, the results of Figure 5J, concerning the enzyme SIRT1, are not mentioned in the text. Therefore, please, make a comment on this result as well.
A5-6: Following the comment, we have updated the mentioned part in the revised version as follows.
The expressions of p-AMPK and SIRT1 in the liver of HFD-fed mice were lower than the NCD mice, whereas GABA and FCLL-GABA supplementation recovered the decreased expressions (Figure 5J-K).
6) Please, report a quantitative analysis of protein expressions of Figures 7C and 8C.
A6. As suggested, we have quantified the protein expressions as follows.
Q7-1) Please correct some mistake like this: the sentence of lines 284-285 appears twice, or, please, change “0.5” to “0.05” in line 189.
A7-1. As suggested, we have corrected the typo error.
Q7-2) Then, please, correctly place the stars of the statistics in Figure 2B, since they appear to refer to the FCLL-1 group rather than the vehicle group or still,
A7-2. For visibility, we enlarged the size of Figure 2B in the revised version and put indicative stars in the correct place. Revised figure 2A is as follows,
Q7-3) please, correct “S-nitrosylation” to “sulfonation” in the caption of Figure 7C. These are just examples; thus, Authors must check thoroughly the text.
A7-2. Following the reviewer’s comment, we have changed S-nitrosylation to “sulfonation” in the revised caption of Figure 7C as follows,
“Figure 7. GABA and fermented Curcuma longa L. extract enriched with GABA diminish ER stress and IRE1α sulfonation of axis in HFD model.”
References
- Gadde, K.M.; Martin, C.K.; Berthoud, H.R.; Heymsfield, S.B. Obesity: Pathophysiology and Management. J Am Coll Cardiol 2018, 71, 69-84, doi:10.1016/j.jacc.2017.11.011.
- Kusminski, C.M.; Bickel, P.E.; Scherer, P.E. Targeting adipose tissue in the treatment of obesity-associated diabetes. Nat Rev Drug Discov 2016, 15, 639-660, doi:10.1038/nrd.2016.75.
- Serra, D.; Mera, P.; Malandrino, M.I.; Mir, J.F.; Herrero, L. Mitochondrial fatty acid oxidation in obesity. Antioxid Redox Signal 2013, 19, 269-284, doi:10.1089/ars.2012.4875.
- Rask-Madsen, C.; King, G.L. Mechanisms of Disease: endothelial dysfunction in insulin resistance and diabetes. Nat Clin Pract Endocrinol Metab 2007, 3, 46-56, doi:10.1038/ncpendmet0366.
- Evans, J.L.; Goldfine, I.D.; Maddux, B.A.; Grodsky, G.M. Are oxidative stress-activated signaling pathways mediators of insulin resistance and beta-cell dysfunction? Diabetes 2003, 52, 1-8, doi:10.2337/diabetes.52.1.1.
- Zeeshan, H.M.; Lee, G.H.; Kim, H.R.; Chae, H.J. Endoplasmic Reticulum Stress and Associated ROS. Int J Mol Sci 2016, 17, 327, doi:10.3390/ijms17030327.
- Bhandary, B.; Marahatta, A.; Kim, H.R.; Chae, H.J. An involvement of oxidative stress in endoplasmic reticulum stress and its associated diseases. Int J Mol Sci 2012, 14, 434-456, doi:10.3390/ijms14010434.
- Holmstrom, K.M.; Finkel, T. Cellular mechanisms and physiological consequences of redox-dependent signalling. Nat Rev Mol Cell Biol 2014, 15, 411-421, doi:10.1038/nrm3801.
- Wu, R.F.; Ma, Z.; Liu, Z.; Terada, L.S. Nox4-derived H2O2 mediates endoplasmic reticulum signaling through local Ras activation. Mol Cell Biol 2010, 30, 3553-3568, doi:10.1128/MCB.01445-09.
- Hourihan, J.M.; Moronetti Mazzeo, L.E.; Fernandez-Cardenas, L.P.; Blackwell, T.K. Cysteine Sulfenylation Directs IRE-1 to Activate the SKN-1/Nrf2 Antioxidant Response. Mol Cell 2016, 63, 553-566, doi:10.1016/j.molcel.2016.07.019.
- Hwang, I.; Jo, K.; Shin, K.C.; Kim, J.I.; Ji, Y.; Park, Y.J.; Park, J.; Jeon, Y.G.; Ka, S.; Suk, S., et al. GABA-stimulated adipose-derived stem cells suppress subcutaneous adipose inflammation in obesity. Proc Natl Acad Sci U S A 2019, 116, 11936-11945, doi:10.1073/pnas.1822067116.
- Tian, J.; Dang, H.N.; Yong, J.; Chui, W.S.; Dizon, M.P.; Yaw, C.K.; Kaufman, D.L. Oral treatment with gamma-aminobutyric acid improves glucose tolerance and insulin sensitivity by inhibiting inflammation in high fat diet-fed mice. Plos One 2011, 6, e25338, doi:10.1371/journal.pone.0025338.
- Untereiner, A.; Abdo, S.; Bhattacharjee, A.; Gohil, H.; Pourasgari, F.; Ibeh, N.; Lai, M.; Batchuluun, B.; Wong, A.; Khuu, N., et al. GABA promotes beta-cell proliferation, but does not overcome impaired glucose homeostasis associated with diet-induced obesity. FASEB J 2019, 33, 3968-3984, doi:10.1096/fj.201801397R.
- Xie, Z.X.; Xia, S.F.; Qiao, Y.; Shi, Y.H.; Le, G.W. Effect of GABA on oxidative stress in the skeletal muscles and plasma free amino acids in mice fed high-fat diet. J Anim Physiol Anim Nutr (Berl) 2015, 99, 492-500, doi:10.1111/jpn.12254.
- Xie, Z.X.; Xia, S.F.; Le, G.W. Gamma-aminobutyric acid improves oxidative stress and function of the thyroid in high-fat diet fed mice. J Funct Foods 2014, 8, 76-86, doi:10.1016/j.jff.2014.03.003.
- Purwana, I.; Zheng, J.; Li, X.; Deurloo, M.; Son, D.O.; Zhang, Z.; Liang, C.; Shen, E.; Tadkase, A.; Feng, Z.P., et al. GABA promotes human beta-cell proliferation and modulates glucose homeostasis. Diabetes 2014, 63, 4197-4205, doi:10.2337/db14-0153.
- Soltani, N.; Qiu, H.; Aleksic, M.; Glinka, Y.; Zhao, F.; Liu, R.; Li, Y.; Zhang, N.; Chakrabarti, R.; Ng, T., et al. GABA exerts protective and regenerative effects on islet beta cells and reverses diabetes. Proc Natl Acad Sci U S A 2011, 108, 11692-11697, doi:10.1073/pnas.1102715108.
- Cavagnini, F.; Pinto, M.; Dubini, A.; Invitti, C.; Cappelletti, G.; Polli, E.E. Effects of gamma aminobutyric acid (GABA) and muscimol on endocrine pancreatic function in man. Metabolism 1982, 31, 73-77.
- Passariello, N.; Giugliano, D.; Torella, R.; Sgambato, S.; Coppola, L.; Frascolla, N. A possible role of gamma-aminobutyric acid in the control of the endocrine pancreas. J Clin Endocrinol Metab 1982, 54, 1145-1149, doi:10.1210/jcem-54-6-1145.
- Furukawa, S.; Fujita, T.; Shimabukuro, M.; Iwaki, M.; Yamada, Y.; Nakajima, Y.; Nakayama, O.; Makishima, M.; Matsuda, M.; Shimomura, I. Increased oxidative stress in obesity and its impact on metabolic syndrome. J Clin Invest 2004, 114, 1752-1761, doi:10.1172/JCI21625.
- Wang, M.; Wey, S.; Zhang, Y.; Ye, R.; Lee, A.S. Role of the unfolded protein response regulator GRP78/BiP in development, cancer, and neurological disorders. Antioxid Redox Signal 2009, 11, 2307-2316, doi:10.1089/ARS.2009.2485.
- Stolz, A.; Wolf, D.H. Endoplasmic reticulum associated protein degradation: a chaperone assisted journey to hell. Biochim Biophys Acta 2010, 1803, 694-705, doi:10.1016/j.bbamcr.2010.02.005.
- Walter, P.; Ron, D. The unfolded protein response: from stress pathway to homeostatic regulation. Science 2011, 334, 1081-1086, doi:10.1126/science.1209038.
- Halliwell, B. Antioxidants in human health and disease. Annu Rev Nutr 1996, 16, 33-50, doi:10.1146/annurev.nu.16.070196.000341.
- Cecarini, V.; Gee, J.; Fioretti, E.; Amici, M.; Angeletti, M.; Eleuteri, A.M.; Keller, J.N. Protein oxidation and cellular homeostasis: Emphasis on metabolism. Biochim Biophys Acta 2007, 1773, 93-104, doi:10.1016/j.bbamcr.2006.08.039.
- Kitada, M.; Koya, D. SIRT1 in Type 2 Diabetes: Mechanisms and Therapeutic Potential. Diabetes Metab J 2013, 37, 315-325, doi:10.4093/dmj.2013.37.5.315.
- Caton, P.W.; Kieswich, J.; Yaqoob, M.M.; Holness, M.J.; Sugden, M.C. Metformin opposes impaired AMPK and SIRT1 function and deleterious changes in core clock protein expression in white adipose tissue of genetically-obese db/db mice. Diabetes Obes Metab 2011, 13, 1097-1104, doi:10.1111/j.1463-1326.2011.01466.x.
- Jeon, S.-M. Regulation and function of AMPK in physiology and diseases. Experimental & Molecular Medicine 2016, 48, e245-e245, doi:10.1038/emm.2016.81.
- Koh, J.-H.; Kim, J.-Y. Role of PGC-1α in the Mitochondrial NAD+ Pool in Metabolic Diseases. International Journal of Molecular Sciences 2021, 22, 4558.
- Weikel, K.A.; Ruderman, N.B.; Cacicedo, J.M. Unraveling the actions of AMP-activated protein kinase in metabolic diseases: Systemic to molecular insights. Metabolism 2016, 65, 634-645, doi:https://doi.org/10.1016/j.metabol.2016.01.005.
- Boonstra, E.; de Kleijn, R.; Colzato, L.S.; Alkemade, A.; Forstmann, B.U.; Nieuwenhuis, S. Neurotransmitters as food supplements: the effects of GABA on brain and behavior. Front Psychol 2015, 6, 1520, doi:10.3389/fpsyg.2015.01520.
- Sahab, N.R.M.; Subroto, E.; Balia, R.L.; Utama, G.L. gamma-Aminobutyric acid found in fermented foods and beverages: current trends. Heliyon 2020, 6, e05526, doi:10.1016/j.heliyon.2020.e05526.
- Ersoy, B.A.; Maner-Smith, K.M.; Li, Y.; Alpertunga, I.; Cohen, D.E. Thioesterase-mediated control of cellular calcium homeostasis enables hepatic ER stress. J Clin Invest 2018, 128, 141-156, doi:10.1172/JCI93123.

Reviewer 3 Report
Journal: Nutrients
Title: GABA and Fermented Curcuma longa L Extract Enriched with GABA Ameliorate Obesity through Nox4-IRE1α Sulfonation-RIDD-SIRT1 Decay Axis in High-Fat Diet-Induced Obese Mice
The authors describe the effect of GABA and fermented Curcuma longa L extract enriched with GABA (FCLL-GABA) in HFD fed mice. They describe clear effects of GABA and FCLL-GABA supplementation. They analysed several aspects of obesity induced changes as weight, fat accumulation, transcription factors, oxidative stress related and lipid metabolism related genes and proteins in different tissues. They show interesting results and overall the experimental setup is well done.
However, there are some points, which need improvement or clarifying.
- English has to be improved. Several phrases are difficult to understand or sound strange. Please give the manuscript to a native English-speaking person to revise it.
- Line 89–93: Do the mice really receive 1 or 2 g/kg GABA? In which form? How (oral gavage described in Fig legend)?
- Line 117-118: GRP78 mentioned twice.
- Line 111, 123, 536-542: Ref Hoang et al 2021 double.
- Line 128-133:Very basic, no ref, no dilution, time and temperature for second AB.
- Line 135-139: What is measured in fixed tissue by DHE? I know only measurement in living cells and cryosections.
- Line 137: (Model no, Company, Country) ?
- Line 141-147: unclear, improve phrasing.
- Line 157-162: mutant SIRT1 (Fig 7 F)?
- Line 165: "Extraction was done with 1 μg of RNA and oligo dT primer kit (insert company and catalog number)" Extraction? Is 1st strand cDNA synthesis meant? Company ...?
- Line 185, 186: Dual-energy X-ray Absorptiometry 206 (DXA) scan description?
- Line 201: 64.417 mg/L GABA confirms not "not observed in water extracted", it is observed.
- Line 202-203: 6883.018 mg proves not "not detected".
- Line 215: Fig 1 A, B, C not clear, please improve.
- Line 220: Fig 2 A, B, C. Accordingly to the representative images in A and C, FCLL-2 mice are much leaner than GABA-1 mice, but this does not correlate with body weight in B, why?
- Line 272: Fig 4 A, B. if the scale bar is 50 um, then the cells in NCD are around 15 um not 50 um. In A cells in HFD vehicle are much larger than FCLL-1, not almost the same as shown in B. Fig 4 G, I, K. In G are the signals for PPAR for GABA-1 weaker than for GABA-2, but in I the bars are almost the same. The same for FAS, the ratio in K does not correlate with the blot G (NCD vehicle to HFD vehicle to GABA-2).
- Line 284-288: The phrase "To investigate on the liver..." is double.
Line 301: higher AMPK levels are not shown, not recognisable in Fig 4 J.
Line 303: Fig 5 A, B, C, D. Bars in graphs look exactly as bars in Fig 4 C, D, E, F. To get same values in eWAT as in liver is very unlikely.
Line 332: Fig 6 A. Is it H&E staining as written in graph or DHE as written in legend? In Fig 6 E same problem as before. Ratio of Nox4 HFD vehicle to FCLL-1 bars the same, in blot FCLL-1 Nox4 stronger than in vehicle and b-actin weaker. - Line 371: Fig 7 A, B. Same problem, ratios of bars in B do not correlate with ratios in blot A for p-IRE, GRP78 and CHOP.
- Line 383 and 405: Why "promote browning of WAT and increasing the activity of the WAT" analysed was BAT.
- Line 406: Fig 8 F, G. Same problem, ratios of bars in G do not correlate with images in F for UPC-1. UPC-1 staining in NCD vehicle much lower than FCLL-2, GABA-1 and GABA-2 in images, stronger as bars imply.
- Line 431: "without disturbing the food intake (Figure 2)." Not shown
Author Response
Rebuttal Letter
Reviewer 3.
The authors describe the effect of GABA and fermented Curcuma longa L extract enriched with GABA (FCLL-GABA) in HFD fed mice. They describe clear effects of GABA and FCLL-GABA supplementation. They analysed several aspects of obesity induced changes as weight, fat accumulation, transcription factors, oxidative stress related and lipid metabolism related genes and proteins in different tissues. They show interesting results and overall the experimental setup is well done.
However, there are some points, which need improvement or clarifying.
Q1. English has to be improved. Several phrases are difficult to understand or sound strange. Please give the manuscript to a native English-speaking person to revise it.
A1. Following the reviewer’s comment, professionals assistance was sought to improve the quality of the manuscript.
Q2. Line 89–93: Do the mice really receive 1 or 2 g/kg GABA? In which form? How (oral gavage described in Fig legend)?
A2. Powdered GABA was dissolved in water and administered daily via oral gavage. Following the comment, we updated the methods section and figure legends as follows,
“Methods: “Reagents (FCLL and GABA in water) and vehicle (water) were administered daily via oral gavage”.
Figure legends: Mice received vehicle or GABA and GABA-enriched Curcuma longa L. extract (1 or 2 g/kg) once daily for 14 weeks by oral gavage”.
Q3. Line 117-118: GRP78 mentioned twice.
A3. Thanks to the comment, we have removed the repeated word.
Q4. Line 111, 123, 536-542: Ref Hoang et al 2021 double.
A4. We have deleted the double-mentioned reference in the revised version.
Q5. Line 128-133:Very basic, no ref, no dilution, time and temperature for second AB.
A5. Following the comment, we have updated the Immunofluorescence method with the necessary information. Updated text as follows,
2.7. Immunofluorescence analysis
“The eWAT was fixed and washed with phosphate-buffered saline (PBS). The sections were exposed to a primary antibody against UCP-1 (sc-518024, Santa Cruz Biotechnologies Inc., CA, USA) for 2 h at room temperature. Following incubation with a primary antibody, the sections were washed thrice for 5 min in PBS. Later, the sections were labeled using anti-mouse IgG- FITC (Sigma, St Louis, MO, USA) for 1 h at room temperature. The sections were washed thrice for 5 min in PBS, and glass coverslips were mounted with 20 μL aqueous-mount solution (Scytek laboratories, USA). Images were captured using an inverted confocal microscope (Leica DMIRE2; Leica Microsystems, Wetzlar, Germany) with a 63× oil immersion objective lens. All the images were captured with the same laser intensities”.
Q6. Line 135-139: What is measured in fixed tissue by DHE? I know only measurement in living cells and cryosections.
A6. Following the comment, we have revised the DHE staining method. Revised DHE staining method as follows
“2.8. DHE staining
Dihydroethidium (DHE) was employed to analyze the intracellular ROS production in WAT. Fresh WAT were fixed in 4% paraformaldehyde on ice for 1 h. The fixed tissues were washed three times by PBS for 10 min on ice before hydrating overnight in 30% sucrose at 4°C. Then, the tissues were infiltrated with OCT (SAKURA, USA) for 2 h, and stored at -80°C. Sections (5 µm) were cut with a freezing microtome (CM3050S; Leica Microsystems). The sections were dried at room temperature for 5 min, and then washed three times by PBS for 5 min. Tissues were exposed to DHE (50 µM, diluted by PBS) at 37 °C for 30 min. All the images were acquired with EVOS M5000 Cell Imaging System (Thermo Fisher Scientific, MA, USA) and relative fluorescence intensities were assessed with ImageJ (National Institutes of Health, Bethesda, MD, USA)”.
Q7. Line 137: (Model no, Company, Country) ?
A7. Following the comment, instrumental information is updated.
Q8. Line 141-147: unclear, improve phrasing.
A8. Thank you for noticing the unclear sentence. Following the comment, we have updated the IRE1α sulfonation method in the revised material and method section. Updated text as follows,
2.9. IRE1α sulfonation assay
“Sulfonation was evaluated as described previously [1]. About ~500 μg WAT was used to obtain WAT lysates. For detection of sulfonation of IRE1α, immunoprecipitation was performed with anti-cysteine-sulfonate (Abcam, ab176487) using lysates and incubated with anti-IRE1α antibody (3294, cell signaling) to detect IRE1α sulfonation. Later, protein A/G Sepharose beads were mixed and incubated for an hour at room temperature. Finally, immunoprecipitates were washed with phosphate-buffered saline (PBS), resolved using SDS-PAGE, and immunoblotted with appropriate antibodies”.
Q9. Line 157-162: mutant SIRT1 (Fig 7 F)?
A9. Following the comment, we have updated the needed information in the revised manuscript. Revised Materials and Methods as follows,
2.11. In vitro IRE1α-mediated SIRT1 cleavage assay
Cleavage assay was performed as described previously [2]. Briefly, a synthesized SIRT1 gene was inserted in plasmid pMA-RQ and linearized with BglII (Promega) to obtain 5’ mRNA fragments containing IRE1 cleavage site (CUGCAG). To generate mutation at the cleavage site at 5’ region of SIRT1 mRNA, pMA-RQ-SIRT1 was cut by ApaI/XhoI, then ApaI-XhoI small fragment (198bp) was removed. The mutant fragment was prepared by PCR with the mutant primer set (SIRT1_ApaI_m1: AGATGGGCCCTACAGGCC or SIRT1_ApaI_m2: AGATGGGCCCTGAAGGCC and SIRT1_XhoI: GGCCTCGAGCGGAGC) and ApaI/XhoI cut. ApaI-XhoI fragment was replaced with the ApaI-XhoI mutant fragment. Mutant sequences were validated by sanger sequencing. T7 transcription kit (AM1333, Thermo Fisher Scientific) was used for in vitro transcription of the 5’ region of SIRT1, and an in vitro cleavage assay of SIRT1 mRNA was done as described earlier [25]. Finally, RNA products were incubated with or without recombinant IRE1 (E31-11G, SignalChem) at 37 °C, and fragments were separated on a 1.5% denaturing agarose gel.
Q10. Line 165: "Extraction was done with 1 μg of RNA and oligo dT primer kit (insert company and catalog number)" Extraction? Is 1st strand cDNA synthesis meant? Company ...?
A10. Following the comment, we have inserted the necessary information. Updated text as follows,
“2.12. Reverse transcription polymerase chain reaction (RT-PCR)
Total RNA from tissue was separated using TRIzol reagent (Invitrogen, Carlsbad, CA, USA). cDNA was synthesized from 1 μg of RNA with PrimeScript reverse transcript reagent Kit (Takara, Tokyo, Japan). Quantitative PCR was done using TaKaRa SYBR premix Ex Taq kit (TaKaRa Bio Inc., Japan) on ABI PRISM 7,500 (Applied Biosystems, Foster City, CA, USA). The comparative cycle threshold (Ct) method was used to quantify the expression levels, and each amplified product was adjusted to β-actin expression. The primer sequences of the genes were designed according to the sequence information from GenBank database (Table 1)”.
Q11. Line 185, 186: Dual-energy X-ray Absorptiometry 206 (DXA) scan description?
A11. Following the comment, we updated the Dual-energy X-ray Absorptiometry (DXA) scan as follows,
2.15. Dual-energy X-ray Absorptiometry (DXA) scan
The percentage of fat and fat mass were measured using a cone-beam flat panel detector DXA (iNSiGHT VET DXA, Osteosys, Korea) according to the manufacturer's instructions. The lean body mass was deduced from the total body weight to calculate fat mass. In color-composition images, fat and lean tissue are indicated in red and green, respectively. To evaluate abdominal fat, the region of interest (ROI) was defined from the whole-body scan. The area indicates a rectangular box extending from one vertebral space to another, with the lateral border extending to the edge of the abdominal tissue. The abdominal fat percentage was calculated using the following equation, abdominal fat (DEXA)/total fat (DEXA) X100.
Q12. Line 201: 64.417 mg/L GABA confirms not "not observed in water extracted", it is observed.
A12. We apologize for the confusion over the sentence. Following the comment, we have updated the description of the quantification. The revised text as follows,
“GABA and FCLL-GABA products were quantified to standardize and confirm the quality of the extracted compound. As illustrated in Figure 1A-B, GABA was identified as a component indicated by the retention time (fermented Curcuma longa L.: 19.642 min, water extracted-Curcuma longa L.:19.600 min). Observations show that GABA in fermented Curcuma longa L. is 8181.385 mg/L, whereas water extracted Curcuma longa L. contained 64.417 mg/L of GABA. These observations clearly show that fermented Curcuma longa L. is enriched with GABA”.
Q13. Line 202-203: 6883.018 mg proves not "not detected".
A13. We appreciate for the comment. We apologize for the unintentional typo error. It was mentioned mistakenly and deleted in the revised manuscript. HPLC data clearly shows that there was no peak. We have updated the text as follows,
“Curcumin was not detected in fermented Curcuma longa L. (16.742 min) and it was detected in water extracted-Curcuma longa L. (16.758 min, 11.322 mg/L) (Figure 1C)”.
Q14. Line 215: Fig 1 A, B, C not clear, please improve.
A14. Following the comment, we updated the Figure Legend as follows:
“Figure 1. HPLC analysis of GABA in fermented- and water extracted-Curcuma longa L. extracts. Representative chromatograms of GABA standards (0, 500, 1,000 and 5,000 ppm) and fermented Curcuma longa L. extracts (A) and water extracted Curcuma longa L. (B). Representative chromatograms of curcumin; Curcuma longa L. water extract and fermented extract (C)”.
Q15. Line 220: Fig 2 A, B, C. Accordingly to the representative images in A and C, FCLL-2 mice are much leaner than GABA-1 mice, but this does not correlate with body weight in B, why?
A15. We appreciate for the valuable comment. We agree with the comment as our representative images (Figure 1B) do not correlate with body weight. Hence, we have deleted the representative images in the revised manuscript to avoid confusion.
Q16. Line 272: Fig 4 A, B. if the scale bar is 50 um, then the cells in NCD are around 15 um not 50 um.
In A cells in HFD vehicle are much larger than FCLL-1, not almost the same as shown in B.
Fig 4 G, I, K. In G are the signals for PPAR for GABA-1 weaker than for GABA-2, but in I the bars are almost the same.
The same for FAS, the ratio in K does not correlate with the blot G (NCD vehicle to HFD vehicle to GABA-2).
A16. We greatly appreciate for the basic but critical comment. During the revision, we quantified the adipocyte size and represented the average adipocyte size (Adipocyte diameter in µm). The updated quantification analysis data (Figures 4A and B) as follows,
Regarding quantification analysis with respect to figure 4G, we re-quantified the PPAR and FAS expression and updated the data in the revised version.
Q17. Line 284-288: The phrase "To investigate on the liver..." is double.
A17. We have removed the repeated text.
Q18. Line 301: higher AMPK levels are not shown, not recognisable in Fig 4 J.
A18. We appreciate the reviewer for the observation. We apologize for the misinterpretation of the AMPK observation. AMPK levels should be similar in all the groups. We have deleted the sentence.
Q19. Line 303: Fig 5 A, B, C, D. Bars in graphs look exactly as bars in Fig 4 C, D, E, F. To get same values in eWAT as in liver is very unlikely.
A19. We apologize for the error during the figure arrangement. We have provided RAW data to review, and we have corrected the error in the revised manuscript.
Raw data as follows,
Figure 5A-D
Q20-1. Line 332: Fig 6 A. Is it H&E staining as written in graph or DHE as written in legend?
A20-1. We apologize for the mistake. We have corrected the mislabeled Fig 6A as follows.
Q20-2. In Fig 6 E same problem as before. Ratio of Nox4 HFD vehicle to FCLL-1 bars the same, in blot FCLL-1 Nox4 stronger than in vehicle and b-actin weaker.
A20-2. We appreciate the reviewer for the valuable comment. We have repeated the same blotting three times to confirm the observation and quantified the intensity ratio based upon beta-actin. We have updated the quantification analysis in the revised manuscript.
Q21. Line 371: Fig 7 A, B. Same problem, ratios of bars in B do not correlate with ratios in blot A for p-IRE, GRP78 and CHOP.
A21. We appreciate the reviewer for the valuable comment. We have repeated the same blotting three times to confirm the observation and quantified the intensity ratio based upon beta-actin. We have updated the quantification analysis in the revised manuscript.
Q22. Line 383 and 405: Why "promote browning of WAT and increasing the activity of the WAT" analysed was BAT.
A22. We apologize for the typo error. Following the comment, we have corrected the error in the revised manuscript. Updated text as follows,
3.7. GABA and fermented Curcuma longa L. extract enriched with GABA regulate lipid metabolism via AMPK-SIRT1 signaling and promote browning of BAT
Q23. Line 406: Fig 8 F, G. Same problem, ratios of bars in G do not correlate with images in F for UPC-1. UPC-1 staining in NCD vehicle much lower than FCLL-2, GABA-1 and GABA-2 in images, stronger as bars imply.
A23. We thank the reviewer for keen observation. We repeated the blotting three times to confirm the observation and re-quantified the data. We updated the quantification data in the revised manuscript.
Q24. Line 431: "without disturbing the food intake (Figure 2)." Not shown
A24. In this investigation, we did not observe the difference in food intake among the groups. Compared to food intake, the presentation of other parameters are critical to the hypothesis. Thus, we did include food intake data. For the reviewer’s information, food intake data is as follows,
References
- Kim, H.K.; Lee, H.Y.; Riaz, T.A.; Bhattarai, K.R.; Chaudhary, M.; Ahn, J.H.; Jeong, J.; Kim, H.R.; Chae, H.J. Chalcone suppresses tumor growth through NOX4-IRE1alpha sulfonation-RIDD-miR-23b axis. Redox Biol 2021, 40, 101853, doi:10.1016/j.redox.2021.101853.
- Lee, H.Y.; Lee, G.H.; Hoang, T.H.; Park, S.A.; Lee, J.; Lim, J.; Sa, S.; Kim, G.E.; Han, J.S.; Kim, J., et al. D-allulose ameliorates hyperglycemia through IRE1alpha sulfonation-RIDD-SIRT1 decay axis in the skeletal muscle. Antioxid Redox Signal 2022, 10.1089/ars.2021.0207, doi:10.1089/ars.2021.0207.

Round 2
Reviewer 2 Report
Authors have fully addressed all the comments made, hence the paper can be now accepted as it is.